# Transcriptional repression of Plxnc1 by Lmx1a and Lmx1b directs topographic dopaminergic circuit formation

Audrey Chabrat[1,2], Guillaume Brisson[1,2], Hélène Doucet-Beaupré[1,2], Charleen Salesse[1,2], Marcos Schaan Profes[1,2], Axelle Dovonou[1,2], Cléophace Akitegetse[1,2], Julien Charest[1,2], Suzanne Lemstra[3], Daniel Côté [2,4], R. Jeroen Pasterkamp [3], Monica I. Abrudan[5,6], Emmanouil Metzakopian[5], Siew-Lan Ang[7] & Martin Lévesque[1,2]

Mesodiencephalic dopamine neurons play central roles in the regulation of a wide range of brain functions, including voluntary movement and behavioral processes. These functions are served by distinct subtypes of mesodiencephalic dopamine neurons located in the substantia nigra pars compacta and the ventral tegmental area, which form the nigrostriatal, mesolimbic, and mesocortical pathways. Until now, mechanisms involved in dopaminergic circuit formation remained largely unknown. Here, we show that Lmx1a, Lmx1b, and Otx2 transcription factors control subtype-specific mesodiencephalic dopamine neurons and their appropriate axon innervation. Our results revealed that the expression of Plxnc1, an axon guidance receptor, is repressed by Lmx1a/b and enhanced by Otx2. We also found that Sema7a/Plxnc1 interactions are responsible for the segregation of nigrostriatal and mesolimbic dopaminergic pathways. These findings identify Lmx1a/b, Otx2, and Plxnc1 as determinants of dopaminergic circuit formation and should assist in engineering mesodiencephalic dopamine neurons capable of regenerating appropriate connections for cell therapy.

[1] Department of Psychiatry and Neurosciences, Faculty of Medicine, Université Laval, Québec, Quebec G1V 0A6, Canada. [2] CERVO Brain Research Centre, 2601, chemin de la Canardière, Québec, Quebec, Canada G1J 2G3. [3] Department of Translational Neuroscience, Brain Center Rudolf Magnus, University Medical Center Utrecht, 3584 CG Utrecht, The Netherlands. [4] Département de Physique, Genie Physique et Optique, Université Laval, Québec, Quebec G1V 0A6, Canada. [5] Wellcome Trust Sanger Institute, Wellcome Trust Genome Campus, Hinxton, Cambridge CB10 1SA, UK. [6] Faculty of Medicine, School of Public Health, Imperial College, London, W2 1PG, UK. [7] The Francis Crick Institute, 1 Midland Road, London, NW1 1AT, UK. Emmanouil Metzakopian, Siew-Lan Ang and Martin Lévesque jointly supervised this work. Correspondence and requests for materials should be addressed to M.L. (email: martin.levesque@fmed.ulaval.ca)

Subsets of midbrain neurons forming the ventral tegmental area (VTA) and the substantia nigra pars compacta (SNpc) produce the majority of dopamine in the central nervous system. Despite sharing a common neurotransmitter phenotype, these neurons innervate different brain regions, serve different functions, and are differentially susceptible to degeneration. In Parkinson's disease (PD), mesodiencephalic dopamine (mDA) neurons from SNpc are the most affected population, whereas VTA neurons degenerate at much later stages of the disease[1]. Dopaminergic axons from SNpc neurons target the dorsal striatum forming the nigrostrial pathway that controls motor behavior. Dopaminergic neurons from the VTA innervate numerous brain structures including the ventral striatal region, the nucleus accumbens, and the prefrontal cortex. VTA neurons are involved in a variety of behavioral processes such as reward and motivation, and they are referred to as the mesolimbic and the mesocortical pathways. Despite the absence of clear boundaries between these two groups of neurons, single axon tracing studies reported that ascending projections from the SNpc and VTA are topographically organized[2–5].

The differential expression of the guidance cue Netrin-1 in the striatum has been shown to contribute to the topographic organization of mDA neuron innervation[6]. However, although some guidance molecules have been shown to influence mDA neurons targeting (see ref. [7] for a review), how the different mDA neuronal subpopulations establish their specific connections remains unclear.

Recent progress has led to the identification of transcription factors expressed in mDA progenitors that are required for their differentiation. Two LIM-homeodomain transcription factors, Lmx1a and Lmx1b (Lmx1a/b), are specifically expressed in mDA progenitors, and this expression persists in post-mitotic and adult mDA neurons. We and others have recently shown that Lmx1a and Lmx1b were required for the survival of adult mDA neurons[8, 9]. Although some evidence suggests that Lmx1a/b might also be involved in axon growth and guidance[10–12], no studies have yet determined their role in the formation of dopaminergic circuits.

Here, we show that Lmx1a and Lmx1b are required for the topographical organization of dopaminergic innervation in the striatum. Our gene expression profiling experiments identified Plxnc1 as a downstream target gene of Lmx1a/b in mature mDA neurons. Using in vivo and in vitro approaches, we found that the interaction of Plxnc1 with its ligand semaphorin 7a (Sema7a) segregates the nigrostriatal and mesolimbic pathways. Moreover, we show that another transcription factor, Otx2, well characterized for its role during mDA neurons development, promotes Plxnc1 expression. Our findings elucidate a central mechanism leading to the establishment of key ascending circuits in the brain and pave the way to the development of a more efficient cell replacement therapy for PD.

## Results

**Lmx1a/b are required for appropriate mDA axon projections.** During midbrain development, Lmx1a/b are among the first transcription factors expressed by mDA progenitors[13, 14]. Previous studies showed that Lmx1a/b are required for the specification, proliferation, and differentiation of dopaminergic progenitors[15, 16]. Immunostaining of mouse embryonic and post-natal midbrain sections shows that all post-mitotic mDA neurons express both Lmx1a and Lmx1b as previously reported (Fig. 1)[16]. Given the functional redundancy established between these two factors in mDA progenitors[16], we generated $Dat^{Cre/+}$; $Lmx1a^{flox/flox}$;$Lmx1b^{flox/flox}$ (referred to henceforth as Lmx1a/b cKO) mice to study potential roles of Lmx1a/b in regulating axon pathfinding of mDA neurons. Lmx1a/b cKO animals were born at the expected Mendelian frequency, fertile and morphologically indistinguishable from their control littermates ($Dat^{+/+}$; $Lmx1a/b^{f/f}$).

In agreement with previous studies[17–19], the Cre recombinase expressed from the Dat promoter is efficient at deleting Lmx1a and Lmx1b in the ventral midbrain (Supplementary Fig. 1). Inactivation of Lmx1a and Lmx1b using the Dat-Cre line starts around embryonic day E14[19], which corresponds to the developmental period where mDA neurons initiate their axonal growth toward the forebrain[7]. Histological analysis and stereological counting of mDA neurons in Lmx1a/b cKO and control mice did not reveal any significant difference in the total number and the distribution of mDA neurons (Fig. 2a–c). However,

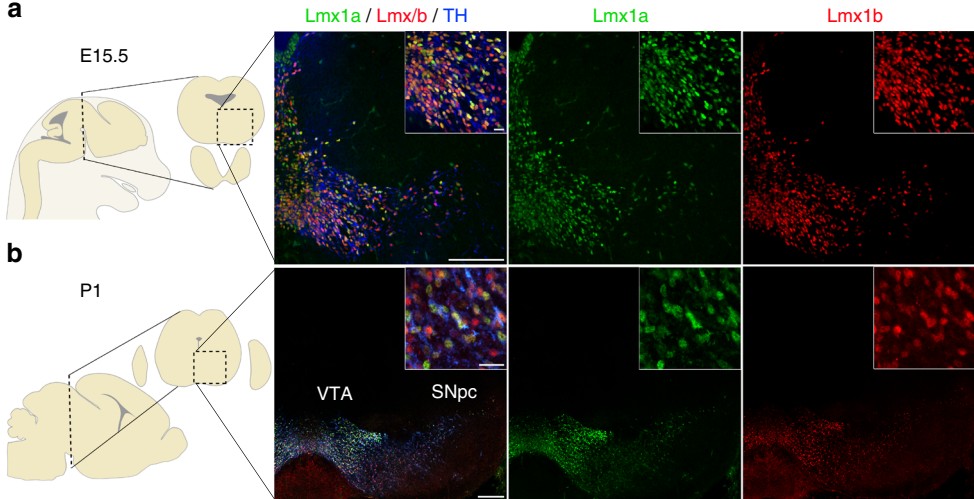

**Fig. 1** Ventral midbrain expression of transcription factors Lmx1a and Lmx1b at early stage of development. Representative confocal stitched images of immunohistochemical staining of the transcription factors Lmx1a and Lmx1b in TH-positive neurons in midbrain coronal sections at E15.5 (**a**) and P1 (**b**). Although Lmx1a and Lmx1b are present in all mDA neurons, the staining intensity for Lmx1a and Lmx1b varies in mDA neurons. Scale bars: **a**, 200 μm and 10 μm for high-magnification inserts; **b**, 200 μm, and 20 μm for high-magnification inserts. SNpc, substantia nigra pars compacta; VTA, ventral tegmental area

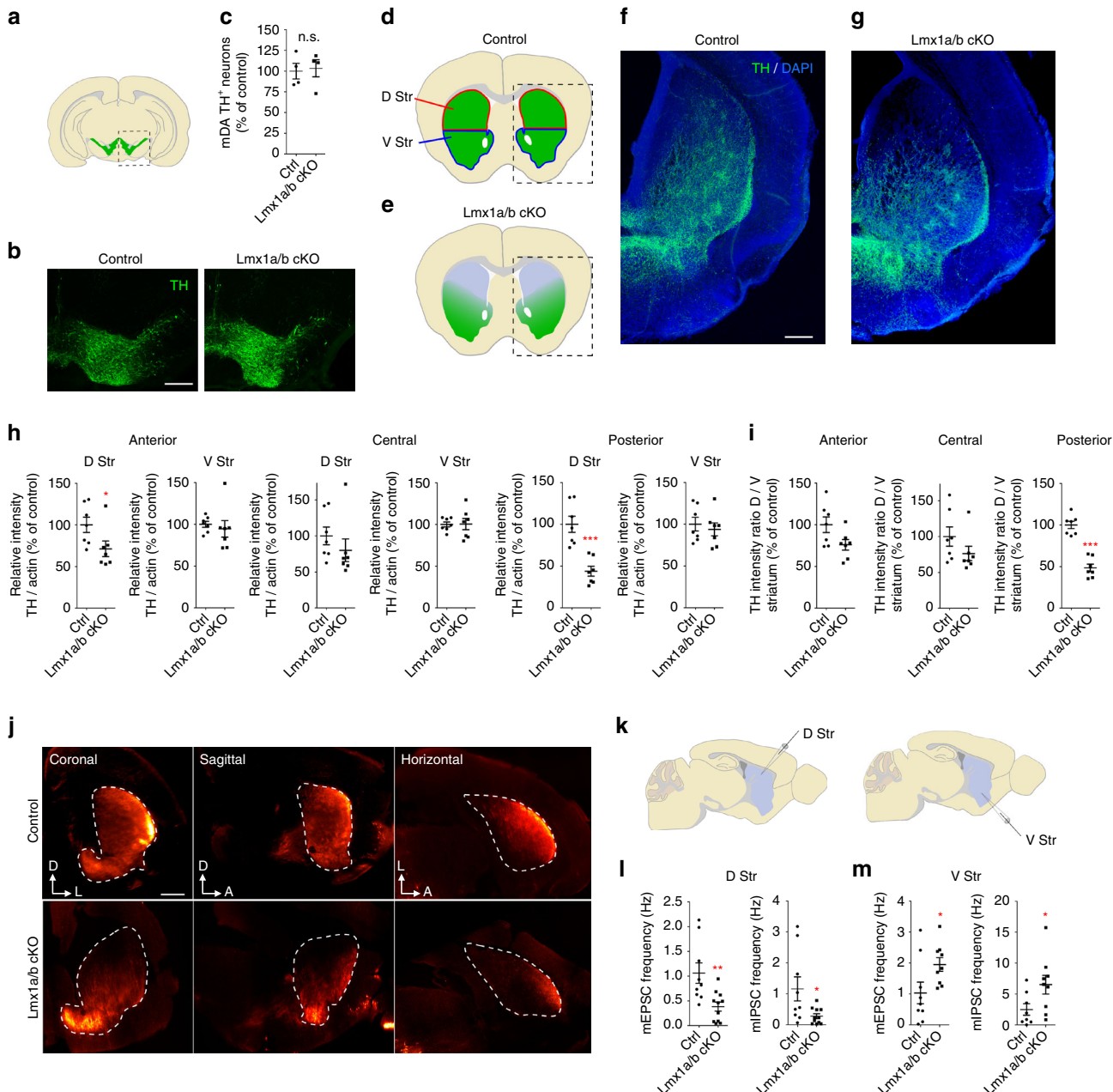

**Fig. 2** Characterization of the phenotype of Lmx1a/b double conditional mutant mice at P1. **a** Schematic representation of coronal section of mouse brain at the midbrain level at post-natal day 1 (P1). Dashed line indicates the delimitation of the pictures shown in **b**. Distribution (**b**) and number (**c**) of mDA neurons in the midbrain were not different between controls and Lmx1a/b cKO at P1 ($n = 4$, two-tailed unpaired $t$ test, $p = 0.8134$). **d, e** Schematic representations of axonal innervation in the striatum in control and Lmx1a/b cKO mice brains at P1. **f, g** Representative confocal images of control and Lmx1a/b cKO mice brain sections showing a loss of dopaminergic innervation in dorso-posterior striatum for Lmx1a/b cKO mice (TH in green and DAPI in blue). **h, i** Optical density measurements of TH axons in the striatum. Graphs in (**i**) show the ratio of TH intensity in dorsal vs. ventral striatum ($n = 7$, two-tailed unpaired $t$ test, $p^{(Ant\ D)} = 0.0483$, $p^{(Cent\ D)} = 0.3398$, $p^{(Cent\ V)} = 0.9596$, $p^{(Post\ D)} = 0.0003$, $p^{(Post\ V)} = 0.5914$, $p^{(Ant\ D/V)} = 0.0526$, $p^{(Cent\ D/V)} = 0.1863$, $p^{(Post\ D/V)} \leq 0.0001$, Mann–Whitney $U$, $p^{(Ant\ V)} = 0.1649$). **j** Light-sheet scans of the TH immunostaining in the striatum of iDISCO-cleared brains from control and Lmx1a/b cKO mice. The panels show single optical planes in coronal and reconstructed sagittal and horizontal planes showing the lack of TH axons in the dorsal striatum of the Lmx1a/b cKO mice brains. Dotted lines delineate the border of the striatum. **k** Schematic representation of the location of electrophysiological recordings in the striatum. **l, m** Analysis of the frequency of miniature excitatory post-synaptic currents (mEPSC) and miniature inhibitory post-synaptic currents (mIPSC) in dorsal and ventral striatum (dorsal striatum: $n = 9$ cells for controls and $n = 11$ cells for Lmx1a/b cKO, Mann–Whitney $U$, $p^{(mEPSC)} = 0.0078$; $p^{(mIPSC)} = 0.0275$; ventral striatum: $n = 9$; two-tailed unpaired $t$ test; $p^{(mEPSC)} = 0.0408$; $p^{(mIPSC)} = 0.0444$). Scale bars: 250 μm for (**b**), (**f**), (**g**) and 500 μm for (**j**). D, dorsal; L, lateral; N.S., not significant; Str, striatum; V, ventral

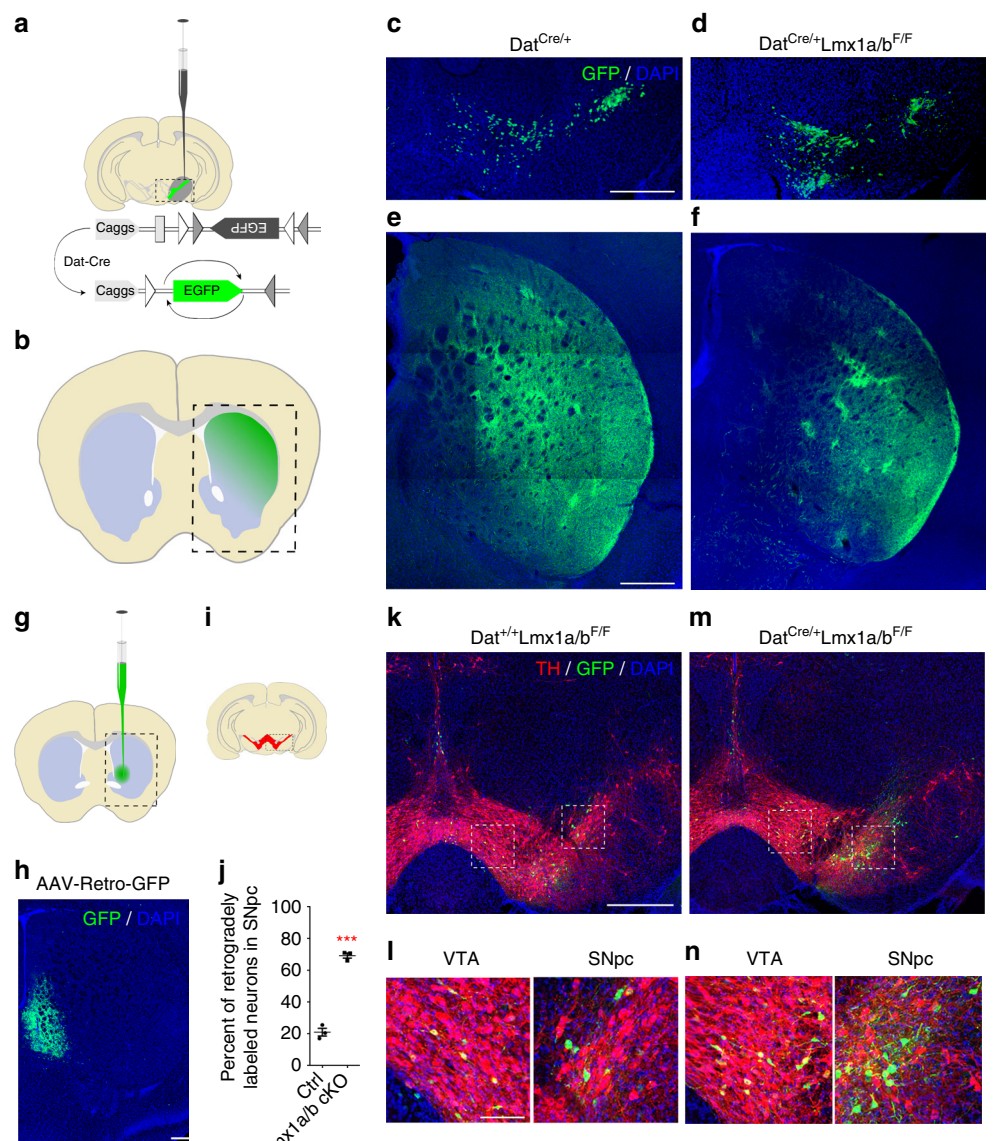

**Fig. 3** Anterograde and retrograde axonal tracing experiments showing aberrant dopaminergic axonal connections in Lmx1a/b conditional mutants. **a**, **b** Schematic description of the AAV-FLEx-EGFP experiment showing injection of the viral vector in the SNpc at P10 and the labeling of the axonal projections in the striatum. **c**, **d** Representative confocal images of the GPF-positive cells in the midbrain (AAV-FLEx-EGFP in green and DAPI in blue), and the resulting GFP-positive axons at the striatal level in the control (**e**) and in the Lmx1a/b cKO mice (**f**) 17 days after injection. **g** Schematic representation of the injection site of the AAV-retro-GFP in the ventral striatum. **h** Representative confocal images of the injection site in the ventral striatum. **i** Schematic representation of a coronal section of mouse brain at the midbrain level indicating the delimitation of the pictures shown in **k**–**n**. **j** Quantification of the percentage of retrogradely labeled neurons in SNpc (GFP$^+$ TH$^+$ in SNpc on total GFP$^+$ TH$^+$; $n = 3$, two-tailed unpaired $t$ test, $p < 0.0001$). **k**–**n** Representative confocal images of the retrogradely labeled cells in control (**k**) and in Lmx1a/b cKO mice (**m**) 17 days after injection in the ventral striatum at P10 (TH in red, AAV-Retro-GFP in green and DAPI in blue). **l**, **n** Higher magnification in the VTA and SNpc as indicated by the dashed boxes in upper images **k**, **m**. Scale bars: **c**, **d**, **e**, **f**, 250 μm; **h**, **k**, **m**, 500 μm; **l**, **n**, 100 μm

analysis of tyrosine hydroxylase (TH) immunostaining in the striatum of Lmx1a/b cKO mice revealed an obvious lack of dopaminergic (DA) axons innervating the dorsal and caudal portion of the striatum at post-natal day 1 (P1) and in 2-week-old mice (Fig. 2d–g and Supplementary Figs. 2 and 3). Accordingly, quantification of TH-positive axonal innervation in the striatum shows a decrease of mDA axon density in the dorso-posterior striatal region (Fig. 2h, i and Supplementary Fig. 2g). No significant defect was observed in other known regions innervated by mDA neurons (Supplementary Fig. 4). We also used iDISCO brain clearing[20] combined with TH immunostaining and light-sheet whole-brain imaging to further analyze the DA system. The

whole-brain analysis of TH immunostaining confirmed the lack of DA axons in the dorso-posterior striatum, but no other obvious defect could be detected in Lmx1a/b cKO mice brains (Fig. 2j).

Since dopamine was shown to induce synapse formation in vitro[21], we evaluated the physiological consequences of a lack of DA axons in dorsal striatum. Post-synaptic currents (PSCs) were recorded in the dorsal and ventral striatum of Lmx1a/b cKO mice and control littermates using whole-cell patch-clamp recordings (Fig. 2k–m). Analysis of miniature inhibitory and excitatory PSC in the dorsal striatum revealed a decrease in the frequency of both excitatory and inhibitory synaptic inputs

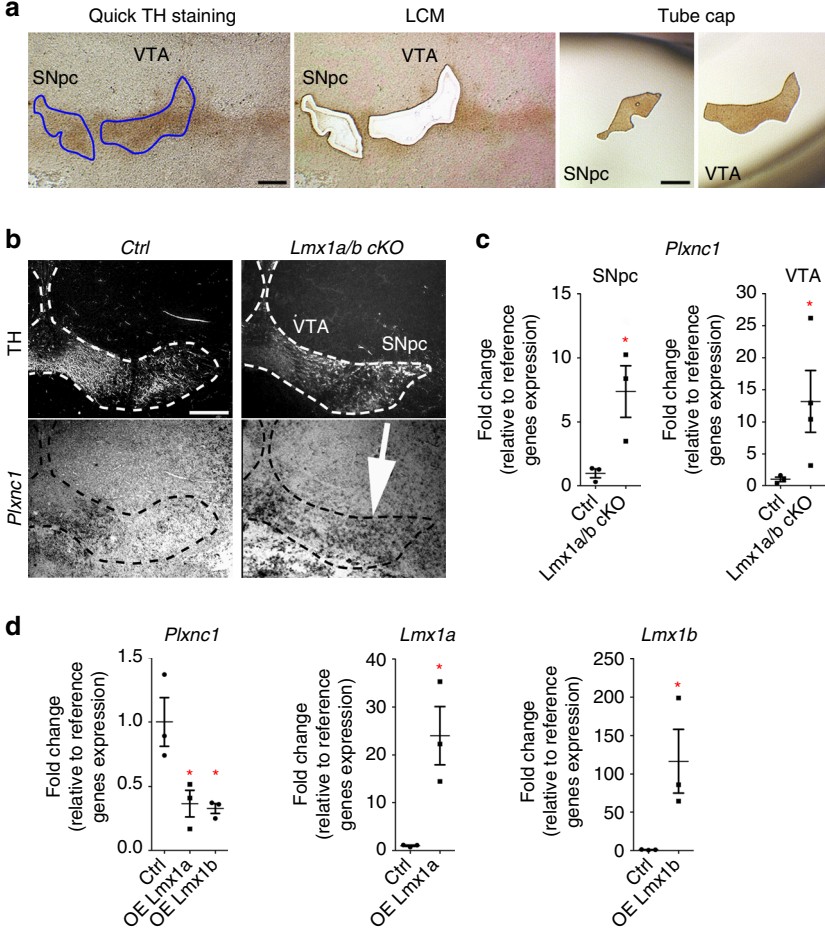

**Fig. 4** Lmx1a/b regulate the expression of the axon guidance receptor Plxnc1. **a** Images of brain tissue section following quick TH staining, before LCM and in the tube cap after LCM. **b** In situ hybridization for *Plxnc1* expression in ventral midbrain coronal sections of control and Lmx1a/b cKO mice at P1. **c** RT-qPCR analysis of *Plxnc1* expression specifically in SNpc and VTA neurons isolated with LCM (SNpc: $n = 3$, two-tailed unpaired $t$ test, $p = 0.0353$; VTA: $n = 3$ controls and 4 Lmx1a/b cKO, one-tailed Mann–Whitney $U$, $p = 0.0286$). **d** RT-qPCR quantification of *Plxnc1* expression after Lmx1a and Lmx1b overexpression in ventral midbrain primary cell cultures at P1 ($n = 3$ independent cultures; for *Plxnc1*: one-way ANOVA with Tukey's post test, *$p < 0.05$; for *Lmx1a* and *Lmx1b*: one-tailed Mann–Whitney $U$, $p = 0.05$). Scale bars: 200 μm

(Fig. 2l). These data are consistent with a lack of DA innervation in the dorsal striatum observed in Lmx1a/b cKO mice. Inversely, we found an increase in the frequency of excitatory and inhibitory synaptic inputs in the ventral striatum of Lmx1a/b cKO mice (Fig. 2m). To confirm a potential axon-targeting defect, we examined specific axon projections of DA neurons from the SNpc and VTA. We performed both anterograde and retrograde tracing experiments. First, we used a Cre recombinase-dependent viral vector expressing green fluorescent protein (GFP) for labeling mDA neurons and their axonal projections[22, 23]. Ten days after birth, Lmx1a/b cKO and control (Dat$^{cre/+}$) mice received stereotaxic injections of AAV-FLEx-GFP in the SNpc (Fig. 3a–f). Upon Cre-mediated recombination, infected DA neurons expressed GFP. Histological analyses showed that in both the controls and Lmx1a/b cKO mice SNpc and part of the VTA neurons expressed GFP following AAV-FLEx-GFP injection (Fig. 3c, d). Analysis of GFP axons innervating the striatum corroborates with the axon-targeting defect observed with TH immunostaining in Lmx1a/b cKO mice. In control animals, GFP axons could be found in the entire striatal region, whereas in Lmx1a/b cKO mice, GFP axons were absent from the dorsal striatal region(Fig. 3e, f). We next confirmed this phenotype by performing stereotaxic injections of a retrograde adeno-associated viral vector (AAV-Retro-GFP[24]) in the ventral striatum

(Fig. 3g–n). When injected in control animals, backfilled neurons were observed predominantly in the VTA (Fig. 3j–l). In contrast, injection of AAV-Retro-GFP in the ventral striatum of Lmx1a/b cKO mice resulted in numerous labeled neurons not only in the VTA but also in the SNpc (Fig. 3m, n). Accordingly, we observed the same result using the retrograde tracer Fluorogold (Supplementary Fig. 5).

**Lmx1a and Lmx1b regulate Plxnc1 expression.** Transcription factors can positively or negatively regulate the expression of hundreds of genes[8, 25, 26]. To identify the genes regulated by Lmx1a/b during the axonal development of mDA neurons, we performed gene expression profiling of Lmx1a/b mutant and control embryos at E15.5. We used laser-capture microdissection (LCM) to isolate TH-immunolabeled regions of the SNpc and VTA from controls and Lmx1a/b cKO mice (Fig. 4a), followed by next-generation RNA sequencing (RNA-seq). We found 225 genes differently expressed between Lmx1a/b cKO and control animals (Supplementary Table 1 and Supplementary Fig. 6). We then searched our RNA-seq list to find genes related to axon development. Among the differently expressed genes, RNA for the axon guidance receptor Plxnc1 was found to be significantly more abundant in Lmx1a/b mutants than in controls. To confirm

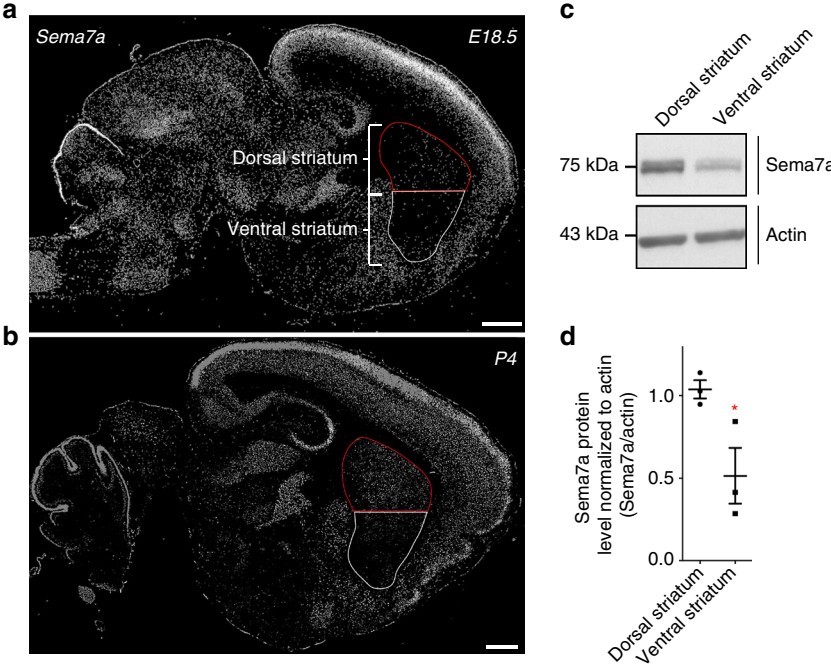

**Fig. 5** The axonal guidance repellent cue Sema7a shows higher expression in the dorsal than in the ventral striatum. **a, b** In situ hybridization for Sema7a on a sagittal section from embryonic day 18.5 and from post-natal day 4 mouse brains (from Allen Brain Atlas). The dorsal striatum (delineated in red) expresses more *Sema7a* than the ventral striatum. **c, d** Western blot and quantification of Sema7a protein level in the dorsal and ventral striatum at P1 ($n = 3$, two-tailed unpaired $t$ test, $p = 0.0418$). Scale bar: 500 μm

the messenger RNA (mRNA) sequencing data, we performed in situ hybridization for *Plxnc1* on midbrain sections at P1 (Fig. 4b). As previously reported for mDA neurons of wild-type mice, *Plxnc1* expression was found restricted to the VTA[27], with no apparent expression detected in the SNpc (Fig. 4b). In contrast, mDA neurons from both the VTA and SNpc expressed significantly higher levels of *Plxnc1* in Lmx1a/b cKO mice (Fig. 4b). Quantification of relative *Plxnc1* mRNA transcript levels in either the SNpc or VTA regions revealed that *Plxnc1* was, respectively, seven and 11-fold more abundant in these two regions in Lmx1a/b mutant mice at P1 (Fig. 4c). Gain-of-function experiments by overexpressing Lmx1a or Lmx1b in ventral midbrain primary cell cultures also induced a significant decrease in *Plxnc1* mRNA levels, confirming the repressive role of Lmx1a/b on *Plxnc1* (Fig. 4d).

**Sema7a/Plxnc1 interaction mediates repulsion of VTA neurons.** Plxnc1 is a known guidance receptor for the membrane-associated GPI (glycosylphosphatidylinositol)-linked semaphorin 7a (Sema7a). Semaphorins are a large family of soluble and membrane bound proteins largely known for their repulsive function on developing axons[28]. In vitro studies have shown that Sema7a binds directly to Plxnc1[29]. However, the functional significance of Sema7a interaction with Plxnc1 expressed by mDA neurons from the VTA remains unknown. Using protein extracts from dorsal and ventral striatum, we quantified the relative abundance of Sema7a protein and found that it was three times more abundant in the dorsal striatum than in the ventral striatum (Fig. 5 and Supplementary Fig. 7). On the basis of the respective level of expression of Plxnc1 in the VTA and Sema7a in the dorsal striatum, we hypothesized that Sema7a acts as a repellent cue on mDA axons from the VTA, and consequently restricts their projections to the ventral striatum. To address this hypothesis, we tested in vitro the effect of Sema7a on ventral midbrain primary cells coming from either the VTA or SNpc. We precisely dissected

ventral midbrain regions containing the SNpc and VTA from 1-day-old Pitx3[Gfp/+] mice and cultured them separately for 2 days. In these mice, all mDA neurons express GFP[30], allowing a more accurate dissection of the VTA and SNpc regions. We then exposed these cultures to Sema7a for 2 h and measured the axonal and dendritic arborization of mDA neurons. Sholl analysis revealed a decrease in both dendritic and axonal complexity of mDA neurons from the VTA but not from the SNpc (Fig. 6a–c). We obtained the same results using VTA and SNpc explants cultured in collagen gel matrix and exposed to Sema7a for 2 h. Both the length and complexity of axons from VTA explants were reduced in the presence of Sema7a, while Sema7a had no effect on mDA axons from SNpc explants (Fig. 6d, e). Interestingly, we also observed a significant enlargement of mDA growth cones from VTA explants following treatment with Sema7a (Fig. 6f, g). Because Sema7a is a membrane-associated GPI-anchored protein and not a diffusible cue, we also tested the axon response of VTA explants grown on alternating stripes of Sema7a. Control stripes had no effect on mDA axons from the VTA as they grew randomly from the explant (Fig. 6h, i). In contrast, mDA axons displayed a clear avoidance for Sema7a stripes when provided with a choice between the control and Sema7a substrate (Fig. 6j, k). We then further assess if the ectopic expression of Plxnc1 observed in SNpc mDA neurons of Lmx1a/b cKO mice has a functional impact on axonal response to Sema7a. We cultured SNpc explants from control and Lmx1a/b cKO mice on alternating stripes of Sema7a and quantified mDA axons terminating on Sema7a. Explants from control SNpc grew randomly on either Sema7a or IgG stripes, confirming that Sema7a has no guidance effect on mDA axons from the SNpc. In contrast, mDA axons from SNpc explants of Lmx1a/b cKO mice displayed a clear avoidance for Sema7a stripes (Supplementary Fig. 8). Altogether, these experiments showed that Sema7a functions as a chemorepulsive and/or non-permissive signal for mDA axons from the VTA. The specific expression of Sema7a

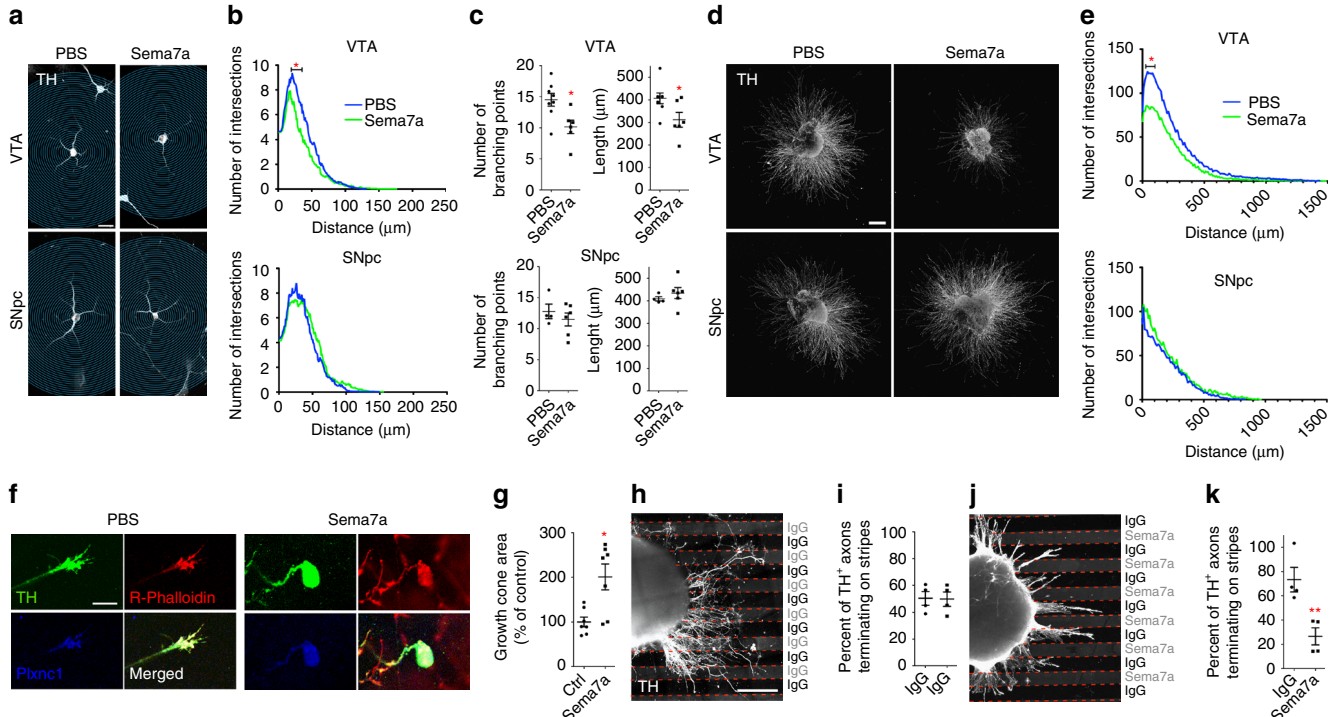

**Fig. 6** In vitro experiments showing the effect of Sema7a on VTA and SNpc neurons. **a**, **c** Sholl analysis on ventral midbrain primary VTA or SNpc neuron cultures exposed to PBS or Sema7a. **a** Examples of confocal images of VTA and SNpc neurons with their respective Sholl intersection circle mask. **b**, **c** Quantitative analysis of the VTA and SNpc neurite intersection profiles, number of branches, and neurite length (SNpc: 71 neurons analyzed from 5 independent cultures; VTA: 114 neurons analyzed from 7 independent cultures; **b**: two-way ANOVA, and the Sidak test was used for post hoc comparisons, *$p < 0.05$; **c**: two-tailed unpaired $t$ test, $p^{(VTA\ Branch)} = 0.0137$, $p^{(VTA\ length)} = 0.0335$, $p^{(SNpc\ Branch)} = 0.4370$, $p^{(SNpc\ length)} = 0.4395$). **d** Examples of embryonic ventral midbrain explants from E14.5 Pitx3-GFP embryos grown in collagen matrix then exposed to PBS (control) or Sema7a. **e** Sholl analysis of the neurite intersection profiles for the VTA and SNpc explants exposed to PBS or Sema7a (a total of 34 explants from 8 independent cultures, 20 explants from VTA and 14 from SNpc; two-way ANOVA, and the Sidak test was used for post hoc comparisons, *$p < 0.05$). **f** Confocal images of growth cones from VTA explants exposed to Sema7a or PBS. **g** Growth cone size measurements expressed as a percentage of control ($n = 7$ independent cultures; 65 growth cones for Sema7a and control were used in each of the 7 experiments, Mann–Whitney $U$, $p = 0.0175$). **h**–**k** Confocal images of stripe assay and quantification of the number of axons terminating on stripes for VTA explants grown on the control alternating IgG stripes and the Sema7a stripes alternating with IgG stripes. Dotted lines delineate the stripes (**i**, $n = 29$ explants in 4 independent experiments, two-tailed unpaired $t$ test, $p = 0.9349$; **k**, $n = 27$ explants in 4 independent experiments, two-tailed unpaired $t$ test, $p = 0.0091$). Scale bars: **a**, 20 μm; **d**, **f**, **h**, 250 μm; **j**, 10 μm

in dorsal striatum may therefore contribute to the dorsal vs. ventral organization of the nigrostriatal and mesolimbic pathways, respectively.

**mDA projections in the striatum is altered in Sema7a KO mice.**
To further study the role of Sema7a on the development of mDA axons, we analyzed the axon projections of mDA neurons in the absence of Sema7a (Fig. 7a–j). Because *Plxnc1* expression is restricted to VTA mDA neurons, loss of Sema7a should affect mDA axon projections to the ventral striatum and nucleus accumbens. As expected, detailed analysis of Sema7a KO mice revealed that mDA axons expressing Plxnc1 were more widely distributed in the dorsal striatum than in control mice (Fig. 7a–d). Quantification of TH and Plxnc1 axonal density in dorsal and ventral striatal regions suggest that TH positive and Plxnc1 positive axons extend more dorsally in Sema7a KO mice (Fig. 7e–g and Supplementary Fig. 9). Stereological counting of mDA neurons in Sema7a KO and control mice did not reveal any significant difference in the total number, the distribution of mDA neurons (Fig. 7h–j), or in the expression of Plxnc1 (Supplementary Fig. 10). To confirm a potential axon-targeting defect of VTA DA neurons in the dorsal striatum, we examined the specific axon projections of DA neurons by injecting an AAV-Retro-GFP into the dorsal

striatal region (Fig. 7k, l). When injected in control animals, backfilled neurons were observed predominantly in the SNpc (Fig. 7m, n). However, injection of AAV-Retro-GFP in the dorsal striatum of Sema7aKO mice resulted in numerous labeled neurons in VTA (Fig. 7o–q). These data suggest that Sema7a in the dorsal striatum functions as a repellent factor for Plxnc1-containing axons.

**Forced expression of Plxnc1 mimics Lmx1a/b cKO phenotype.**
Our in vitro and in vivo experiments suggest that the interaction of Sema7a with Plxnc1 could be responsible for the topographical segregation between the two main DA circuits: the nigrostriatal and mesolimbic pathways. To determine whether the aberrant Plxnc1 expression observed in SNpc neurons of the Lmx1a/b cKO mice could lead to axon targeting defects, we generated a transgenic mouse in which Plxnc1 is produced by all DA neurons. These animals were generated by pronuclear injection of a plasmid in which the TH promoter drives Plxnc1 and td-Tomato expression (TH-Plxnc1-Ires-tdTomato; Fig. 7r–x). In these mice, the expression of the td-Tomato reporter gene was visualized in virtually all mDA neurons (Fig. 7r, s). We also measured the expression level of Plxnc1 by RT-qPCR from 1-mm thick ventral midbrain sections and found that these mice showed a more than fivefold increase of the normal level of Plxnc1 expression (Fig. 7t).

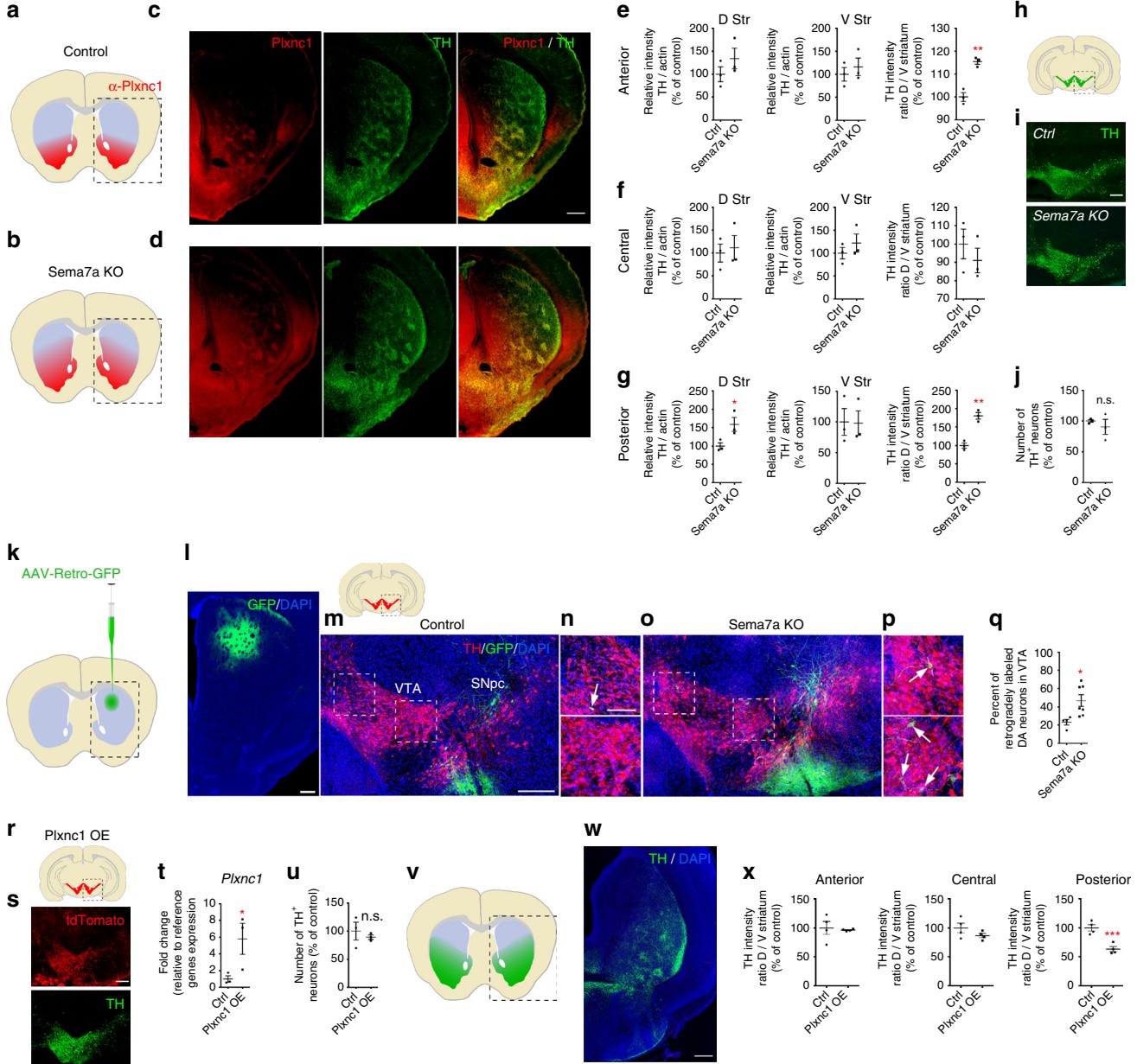

**Fig. 7** Sema7a/Plxnc1 interaction regulates mDA axon guidance. **a**, **b** Schematic representations of coronal views of the axonal projections in the striatum showing that Plxnc1 mDA axons (in red) are more numerous in the dorso-posterior striatal region of Sema7a KO mice (**b**) compared to control animals (**a**). **c**, **d** Confocal images of TH and Plxnc1 immunostained striatal sections from control and Sema7a KO mice. **e–g** Quantification of TH relative optical density at different striatum levels ($n = 3$, two-tailed unpaired $t$ test, $p^{(Ant\ D)} = 0.2855$, $p^{(Ant\ V)} = 0.5355$, $p^{(Cent\ D)} = 0.7461$, $p^{(Cent\ V)} = 0.3933$, $p^{(Post\ D)} = 0.0491$, $p^{(Post\ V)} = 0.9514$, $p^{(Ant\ D/V)} = 0.0023$, $p^{(Cent\ D/V)} = 0.4418$, $p^{(Post\ D/V)} = 0.0023$). **h** Schematic representation of coronal section of mouse brain at the midbrain level. Dashed line indicates the boundaries of the pictures shown in **i**. **i** Representative confocal images of the VTA and SNpc for Control and Sema7a KO mice. **j** Stereological counts of mDA neurons in control and Sema7a KO mice at P1 ($n = 3$, two-tailed unpaired $t$ test, $p = 0.4843$). **k** Schematic representation of the injection site of the AAV-retro-GFP in the dorsal striatum. **l** Representative confocal images of the injection site in the dorsal striatum. **m–p** Representative confocal images of the retrogradely labeled cells in control (**m**, **n**) and in Sema7a KO mice (**o**, **p**) 17 days after injection in the dorsal striatum at P30 (TH in red, AAV-Retro-GFP in green and DAPI in blue). **n**, **p** Higher magnifications in the VTA as indicated by the dashed boxes in **m**, **o**. **q** Quantification of the percentage of retrogradely labeled neurons in VTA (GFP⁺ TH⁺ in VTA on total GFP⁺ TH⁺; $n = 4$ for controls and $n = 7$ for Sema7a KO mice; two-tailed unpaired $t$ test, $p = 0.0245$). **r**, **s** Schematic and confocal images of TH labeling in Plxnc1 overexpression mice showing a loss of DA innervation in dorso-posterior striatum. **t** RT-qPCR quantification of Plxnc1 from the ventral midbrain section of control and Plxnc1 overexpression mice at P1 ($n = 3$, one-tailed unpaired $t$ test, $p = 0.0322$). **u** Stereological counts of mDA neurons in control and Plxnc1 overexpression mice at P1 ($n = 3$, two-tailed unpaired $t$ test, $p = 0.5431$). **v**, **w** Schematic and confocal images of TH and td-Tomato in the VTA and SNpc for one hemisphere. The reporter tdTomato indicates that all mDA neurons express the transgene (Plxnc1-ires-tdTomato). **x** Optical density measurements of TH-positive axons in the striatum at three antero–posterior levels ($n = 4$, two-tailed unpaired $t$ test, $n = 4$, Mann–Whitney $U$, $p^{(Ant\ D/V)} = 0.6857$, two-tailed unpaired $t$ test, $p^{(Cent\ D/V)} = 0.1945$, $p^{(Post\ D/V)} = 0.0019$). Scale bars: 200 μm except **n**, **p**, 100 μm

Stereological counting of mDA neurons labeled for TH and td-Tomato showed no difference in the total mDA neuron number or distribution (Fig. 7u). However, analysis of DA axon projections of these mutant animals revealed a very similar axon-targeting defect to the one observed in Lmx1a/b cKO mice (Figs. 2, 7v, w). The density of DA axons innervating the dorsal and posterior striatal region of these mutants was also reduced

(Fig. 7x), thus indicating that the nigrostriatal circuits were defective in these mutants.

**Otx2 and Lmx1a/b regulate Plxnc1 expression in mDA neurons.** Among the transcription factors expressed by mDA neurons, very few have a restricted localization in either the VTA

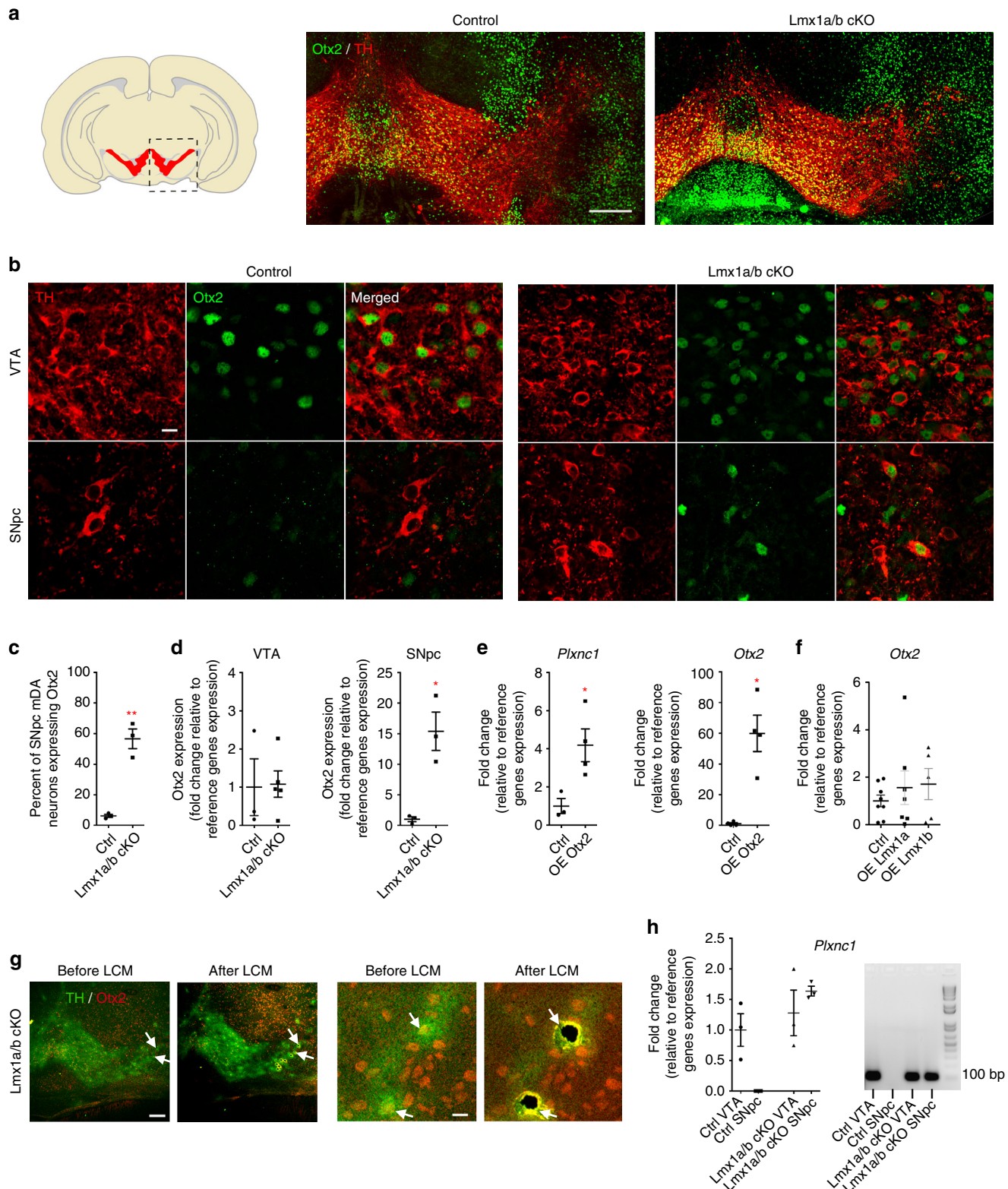

or SNpc[31]. The transcription factor Otx2 is restricted to VTA neurons and has been shown to regulate the subtype identity of mDA neurons[32, 33]. In contrast to Otx2, Lmx1a and Lmx1b are found in both SNpc and VTA neurons. Our results show that Lmx1a/b are repressing Plxnc1, but because Plxnc1 remains expressed in the VTA, we also determined whether Otx2 could contribute to the regulation of *Plxnc1* expression in the VTA. First, we performed double immunolabeling to determine whether Otx2 could also be ectopically expressed in the SNpc of Lmx1a/b cKO mice. As previously reported, Otx2 was restricted to DA neurons of the VTA in control mice[34, 35]. However, we observed numerous DA neurons co-expressing Otx2 and TH in the SNpc of Lmx1a/b cKO mice at P1 (Fig. 8a–c). We next validated if Otx2 expression is upregulated in Lmx1a/b cKO by quantifying Otx2 expression in the VTA and SNpc samples obtained by LCM at P1. We found that Otx2 expression was significantly higher in the SNpc of Lmx1a/b cKO mice compared to the control animals (Fig. 8d). However, Otx2 transcript levels were not significantly changed in the VTA of Lmx1a/b cKO animals. We then overexpressed Otx2 using transient transfection in ventral midbrain primary cell cultures and measured Plxnc1 expression by RT-qPCR. Overexpression of Otx2 induced a significant increase of Plxnc1 transcript levels (Fig. 8e). Finally, to understand the mechanisms controlling subtype identity in mDA neurons, we overexpressed Lmx1a and Lmx1b in ventral midbrain primary cell cultures and measured Plxnc1 and Otx2 expression by RT-qPCR. As predicted, Lmx1a and Lmx1b overexpression led to a decrease in Plxnc1 transcripts (Fig. 4e). However, Lmx1a and Lmx1b overexpression seemed not to be sufficient to repress Otx2 (Fig. 8f). To test if the ectopic Otx2 + mDA neurons in SNpc of Lmx1a/b cKO mice express Plxnc1, we performed rapid double immunolabeling for TH and Otx2, followed by LCM to isolate double-labeled neurons (Fig. 8g). RT-qPCR quantification revealed that similarly to VTA mDA neurons, Otx2- and TH-expressing neurons in SNpc of Lmx1a/b cKO mice also express Plxnc1 (Fig. 8h). Remarkably, we could not detect Plxnc1 expression in mDA neurons of SNpc from control animals. Altogether these results indicate that *Plxnc1* expression in the VTA is maintained by the transcriptional control of Otx2, while in SNpc, Lmx1a and Lmx1b repress *Plxnc1* expression. Consequently, the specific topography of VTA and SNpc axon projections to the striatum appears to be established by the presence or the absence of Plxnc1 in mDA axons.

## Discussion

The proper functioning of the brain's DA system depends on the precise organization of DA circuits during development. Here, we show that transcriptional regulation of Plxnc1 by Lmx1a/b and Otx2 controls the formation and the topographical organization of the nigrostriatal and mesolimbic pathways. We found that the

inactivation of Lmx1a/b in post-mitotic mDA neurons leads to aberrant nigrostriatal axon projections. We also discovered that Lmx1a/b function as a transcriptional repressor (directly or indirectly) of *Plxnc1* expression. Our in vitro data and mutant mice analysis revealed a new mechanism whereby the interaction of Sema7a/Plxnc1 regulates the striatal innervation.

In addition to their roles in the early development of mDA progenitors[15, 16], we showed here that Lmx1a and Lmx1b have other functions in post-mitotic mDA neurons. By controlling Plxnc1 expression, Lmx1a, Lmx1b, and Otx2 regulate the proper organization of nigrostriatal and mesolimbic axon projections. In SNpc, Lmx1a and Lmx1b are required and sufficient to repress Plxnc1. Lmx1a and Lmx1b are also required for the repression of Otx2 in the SNpc, but were not sufficient to repress Otx2 in primary ventral midbrain cell cultures. In the VTA, Lmx1a/b are required and sufficient for Plxnc1 repression, but they are not required or sufficient to repress Otx2. Our gain-of-function experiments also showed that Otx2 can drive Plxnc1 expression in mDA primary cultures. Altogether, our data indicate that Plxnc1 expression is precisely controlled in mDA neurons by the coordinated action of Lmx1a, Lmx1b, and Otx2. In the VTA, all three transcription factors are present. Lmx1a/b repress Plxnc1, but Otx2 promotes and possibly maintains Plxnc1 expression. In the SNpc, Otx2 is normally absent, and both Lmx1a and Lmx1b repress Plxnc1. Following Lmx1a/b ablation, we found ectopic Otx2 expression in mDA neurons located in the SNpc and these neurons also express Plxnc1. In addition, RT-qPCR experiments in Lmx1a/b cKO mice revealed that Otx2 expression was only increased in the SNpc, whereas in the VTA, Otx2 expression was unchanged. According to these results, it is likely that another transcription factor, only present in the SNpc, cooperatively contributes to Otx2 repression in the SNpc.

Previous anatomical studies described the expression of Sema7a in the striatum and Plxnc1 in the VTA, thus suggesting a possible role of these proteins in the guidance of mDA neurons[7, 27]. We showed here that Sema7a is more abundant in the dorsal compared to the ventral striatal region and found that Sema7a repels Plxnc1-containing mDA axons from the VTA. In Sema7a knockout mutant mice, Plxnc1 axons extend more dorsally in the striatum, whereas transgenic mice overexpressing Plxnc1 in mDA neurons lack DA innervations in the dorsal striatum. These in vivo data indicate that the Sema7a/Plxnc1 interaction controls DA axon targeting in the striatum. In addition, our in vitro assays revealed that Sema7a functions as a repellent cue for VTA axons expressing Plxnc1. The repellent function of Sema7a on mDA axons from the VTA reported here represents a major mechanism assuring the proper topographical innervation of the main mDA target region, the striatum. It is reasonable to think that other axon guidance cues contribute to attract mDA neurons in the striatum. Indeed, a gradient of Netrin-1 is present in the striatum. The latero-ventral region

**Fig. 8** Otx2 and Lmx1a/b control Plxnc1 expression in the VTA. **a, b** Confocal images of Otx2 immunostaining in TH-positive cells in coronal midbrain sections of control and Lmx1a/b cKO mutant mice showing ectopic Otx2 expression in SNpc mDA neurons. **c** Quantification of the number of mDA neurons in SNpc with ectopic expression of Otx2 ($n = 3$, two-tailed unpaired $t$ test, $p = 0.0015$). **d** Rt-qPCR quantification of Otx2 in VTA and SNpc isolated from control and Lmx1a/b cKO with LCM (VTA: $n = 3$ for control and $n = 5$ for Lmx1a/b cKO, two-tailed unpaired $t$ test, $p = 0.9128$; in SNpc $n = 3$, one-tailed Mann–Whitney U, $p = 0.05$). **e** Overexpression of Otx2 in primary cell cultures leads to an increase of Plxnc1 expression as quantified by RT-qPCR ($n = 3$ independent cultures; for Plxnc1: two-tailed unpaired $t$ test, $p = 0.03$; for Otx2: Mann–Whitney U, $p = 0.0286$). **f** RT-qPCR quantification of Otx2 in mDA primary cell cultures following Lmx1a and Lmx1b overexpression ($n = 8$ for control, $n = 7$ for Lmx1a and $n = 5$ for Lmx1b; one-way ANOVA with Tukey's multiple comparison test, not significant). **g** Representative images of TH and Otx2 immunolabeling from E15.5 Lmx1a/b cKO embryo at E15.5 showing SNpc neurons co-expressing Otx2 and TH (arrows) before and after LCM. Right panels are high magnification of the left panels. **h** RT-qPCR quantification of Plxnc1 expression from LCM isolated cells co-expressing Otx2 and TH in VTA and SNpc from control and Lmx1a/b cKO embryos. For control SNpc, TH-positive cells were isolated ($n = 3$, 5–8 cells per embryo, one-way ANOVA with Tukey's multiple comparison test, not significant between groups expressing Plxnc1). Image on the right is a representative image of the gel showing the RT-qPCR product. Plxnc1 was not detected in mDA neurons of SNpc from control. Scale bars: **a**, 200 μm; **b**, 10 μm; **g** (left), 100 μm, (right), 10 μm

**Table 1 Antibodies used for this study**

| | Antibodies | Concentration | Origin | Catalog number | Reference |
|---|---|---|---|---|---|
| Primary antibodies | Rabbit anti-TH | 1/1000 (IHC) | Pel-FreezBiologicals | P40101-0 | 1 |
| | Sheep anti-TH | 1/1000 (IHC) | Millipore | AB1542 | 2 |
| | Rat anti-DAT | 1/500 (IHC) | Millipore | MAB369 | 3 |
| | Sheep anti-Plxnc1 | 1/150 (IHC) | R&D systems | AF5375 | 4 |
| | Rabbit anti-Lmx1a | 1/1000 (IHC) | Millipore | AB10533 | 5 |
| | Guinea-pig anti-Lmx1b | 1/100 (IHC) | Dr. Carmen Birchmeier, Max Delbruck Center of Molecular Medicine, Berlin, Germany | | 6 |
| | Goat anti-Otx2 | 1/200 (IHC) | R&D systems | AF1979 | 7 |
| | Mouse anti-Tyrosine tubuline | 1/3000 (IHC) | Sigma | T9028 | 8 |
| | Mouse anti-Actin | 1/1000 (IHC) 1/10,000 (WB) | Millipore | MAB1501 | 9 |
| | Chicken anti-GFP | 1/500 (IHC) | Aves Labs | GFP-1020 | 10 |
| | Rabbit anti-Sema7a | 1/500 (WB) | Abcam | ab23578 | 11 |
| Secondary antibodies | Donkey anti-mouse Alexa-Fluor-680 | 1/5000 (IHC) | Jackson ImmunoResearch | 715-625-150 | |
| | Donkey anti-sheep Alexa-Fluor-680 | 1/5000 (IHC) | Jackson ImmunoResearch | 713-625-147 | 12 |
| | Donkey anti-rabbit Alexa-Fluor-790 | 1/5000 (IHC) | Jackson ImmunoResearch | 711-655-152 | 13 |
| | Donkey anti-mouse Alexa-Fluor-790 | 1/5000 (IHC) | Jackson ImmunoResearch | 715-655-150 | |
| | Donkey Alexa-Fluor-488 | 1/200 (IHC) | Jackson ImmunoResearch | | |
| | Donkey Alexa-Fluor-555 | 1/200 (IHC) | Jackson ImmunoResearch | | |
| | Donkey Alexa-Fluor-647 | 1/200 (IHC) | Jackson ImmunoResearch | | 14 |
| | HRP-conjugated Goat anti-mouse | 1/5000 (WB) | Life Technologies | G-21040 | |
| | HRP-conjugated Goat anti-rabbit | 1/3000 (WB) | Cell Signaling Technology | 7074P2 | |

of the striatum is rich in Netrin-1, whereas the medio-dorsal striatum is poor in Netrin-1. Although all mDA neurons express the receptor DCC[36], VTA axons prefer a higher Netrin-1 concentration than the SNpc axon[6]. Accordingly, null mice for Netrin-1 show a loss of innervation in the dorsal striatum[6]. These data suggest that mDA neurons require both Sema7a and Netrin-1 cues to correctly innervate their striatal targets. SNpc neurons, which do not express Plxnc1, are not repelled by the high concentration of Sema7a in the dorsal striatum, but require Netrin-1 to attract them toward the dorsal striatum. The mechanism by which Netrin-1 differently attracts VTA and SNpc DA axons is not fully understood, but the convergence of guidance signals could direct the trajectory of mDA neurons. A recent study showed that motor growth cones synergistically integrate both Netrin-1 and Ephrin signals. Receptors for these ligands can form a complex that acts in synergy on a common downstream effector[37]. In the striatum, Sema7a and Netrin-1 could also synergistically interact at the receptor level and control mDA innervation in the striatum. In Lmx1a/b cKO mice, we found SNpc axons redirected ventrally. Although our electrophysiological recordings suggested an increase in DA in the ventral striatum, we did not observe a significant increase in DA axon density in this region. One possible explanation is that SNpc axonal branching is limited by another cue such as the high Netrin-1 concentration. It is also possible that although SNpc axons are repelled ventrally, they compete with VTA axons for the synaptic space, which in turn limits the number of SNpc axons ventrally.

The mechanism regulating DA innervation in the striatum reported here sheds new light on how the VTA and SNpc establish their connections with the striatum. However, anatomical and genetic evidence indicate that mDA neurons can be subdivided into more distinct subpopulations[2, 3, 38, 39]. In the striatum, subregions can be divided based on their neurochemical content and two main compartments can be defined, the striosomes and the surrounding matrix[40]. In the SNpc, neurons from the ventral tier are believed to preferentially innervate the striosomes, whereas the ones from the dorsal tier preferentially innervate the matrix compartment[38]. In our study, we did not observe any changes in the striosomes and matrix innervation in either Lmx1a/b cKO, Sema7 KO, or Plxnc1 overexpressing mice. These data suggest that the specific DA projections toward these striatal compartments are regulated by other developmental mechanisms. On the basis of gene expression, multiple molecularly distinct subgroups of mDA neurons have been identified[39], but the precise axon projections of these subpopulations and the guidance mechanisms regulating these subpopulations remain to be discovered.

Our findings have important implications for stem cell engineering and transplantation for PD. This chronic progressive neurodegenerative disorder is characterized by the selective loss of DA-containing neurons in the SNpc. The identification of Plxnc1 as a main guidance receptor allowing subset-specific axon targeting of mDA neurons provides a valuable tool to improve the efficiency of reconnection of newly generated neuron grafts. Indeed, to replace degenerated SNpc mDA neurons that will innervate and integrate specifically in the dorsal striatum, newly generated neurons should not express Plxnc1. In a recent study, forced expression of Otx2 has been used in human embryonic stem cells (hESCs) to generate DA neurons. When grafted in the SNpc, these hESC-derived DA neurons were able to innervate the forebrain structures, but displayed a strong preference for

VTA-specific target regions with few axons innervating the dorsal striatum[41]. According to our results, Otx2 promotes Plxnc1 expression, and thus it is likely that the grafted neurons in this study innervated VTA target regions because they express Plxnc1. Further studies are needed to test the efficiency of a cell replacement therapy using mDA neurons that lack Plxnc1 expression.

In sum, our work shows a new mechanism for Lmx1a and Lmx1b during mid-gestational stages to control the DA innervation of the striatum. Other mechanisms involved in the development of the dopamine circuits still remain largely unknown, despite the involvement of these circuits in important physiological functions and in various mental disorders such as schizophrenia. Interestingly, three SNIPs in LMX1A and one in LMX1B were found associated with schizophrenia, and were the same as those previously identified in PD[42]. However, further studies will be necessary to evaluate the role of these SNIPs in regulating Plxnc1 expression and the development of the DA system.

## Methods

**Animals.** All animal experiments were performed in accordance with the Canadian Guide for the Care and Use of Laboratory Animals and were approved by the Université Laval Animal Protection Committee. $Lmx1a^{f/f9, 15}$, $Lmx1b^{f/f43}$, $Dat^{Cre/+18}$, $Pitx3^{Gfp/Gfp30}$, and Sema7a KO[28] mice were genotyped as previously described. $Lmx1a/b$ cKO mice were generated by intercrossing $Dat^{Cre/+};Lmx1a^{f/f};$ $Lmx1b^{f/f}$ male and $Lmx1a^{fl/fl};Lmx1b^{fl/fl}$ female animals. Plxnc1-Ires-td-Tomato mice were generated by standard procedures using pronuclear microinjection into fertilized single-cell mouse embryos as previously described[44]. Transgenes contained the 9-Kb TH promoter[45] (gift from Dr. Kazuto Kobayashi), the full coding sequence of mouse Plxnc1 (NM_018797.2), and td-Tomato (Plateforme d'outils moléculaires, IUSMQ). Transgenic mice were identified by PCR with forward primer in Plxnc1 sequence, 5′–GGCTGGAAGAAGCTCAGAAA–3′; and reverse primer in IRES sequence, 5′–TACGCTTGAGGAGAGCCATT–3′. Sema7a KO and Plxnc1-Ires-td-Tomato mice are kept in C57BL/6 background, while all other mice were kept in a mixed genetic background. The number of mice used in data quantifications was at least three per groups. No mice were excluded from our analysis. No randomization was applied.

**Tissue analysis.** Mouse brains at P1 were incubated in 4% paraformaldehyde in PBS at 4 °C, followed by cryoprotection in 30% sucrose in PBS, before freezing on dry ice. For mice older than P1, perfusion using 4% paraformaldehyde in PBS was instead performed. After cryostat sectioning at 60 μm, sections were washed in PBS, then blocked with 1% normal donkey serum (NDS) and 0.2% Triton X-100 for at least 30 min. See Table 1 for antibodies used in this study.

**Optical density measurements on sections.** Striatal sections were co-labeled with TH and actin or with Plxnc1 and actin, and then incubated with infrared fluorescent conjugated secondary antibodies (donkey anti-mouse Alexa-Fluor-680, donkey anti-mouse Alexa-Fluor-790, donkey anti-sheep Alexa-Fluor-680, donkey anti-rabbit Alexa-Fluor-790, 1:5000, Jackson ImmunoResearch). Sections were then mounted on glass slides, coverslipped, and scanned using an infrared imaging system (Odyssey CLx; LI-COR Biosciences, Lincoln, NE, USA[46, 47]. Optical density measurements were done using Image Studio Lite Ver 5.2 software after delineating the striatal areas with the software's freehand tool. Although there is no clear boundaries between SNpc and VTA target regions in the striatum, the dorsal half of the striatum receive mostly DA projections from SNpc, whereas the ventral half receives mostly projections from VTA. The distinction between dorsal vs. ventral striatum areas was thus determined by dividing the striatum in two halves, which represent the major striatal target regions for SNpc and VTA[2, 3, 5, 48].

**In situ hybridization.** In situ hybridization was performed as previously described[8, 16]. Brains were dissected in diethylpyrocarbonate (DEPC)-treated PBS, and fixed in 4% PFA (DEPC) overnight. After an overnight incubation in 30% sucrose (DEPC), brains were frozen with dry ice and then cryosectioned in 20-μm thick coronal sections. Sections were collected on superfrost plus slides (Fisher Scientific). The complementary DNA (cDNA) templates were generated from E18.5 whole-brain RNA and used to generate RNA probes for Plxnc1, Lmx1a, and Lmx1b using RT-PCR following the protocol described previously[11].

**Quick TH staining and LCM.** The experiment was performed as described in ref.[49]. Briefly, E15.5 or P1 mouse brains were quickly dissected and snap-frozen in liquid nitrogen. The 12-μm thick cryostat sections were collected on membrane-coated glass-slides (Leica Biosystems), allowed to dry, before being fixed in 70% ethanol at −20 °C. The fixed sections were quickly stained (20 min) using rabbit

anti-TH (Pel-Freez, 1:25) as a first antibody, washed with PBS, and then exposed to a biotinylated anti-rabbit secondary antibody (Vector Labs, 1:100).The slides were subsequently incubated in avidin/biotinylated enzyme complex HRP (Vectastain; Vector Laboratories), and the staining was detected with the diaminobenzidine substrate. The stained slides were stored frozen at −80 °C. After defrosting, the SNpc and VTA areas were collected using a LCM microscope (Leica). For single-cell LCM experiment, slides were co-labeled using a rabbit anti-TH and a goat anti-Otx2, and then exposed to secondary antibodies donkey anti-rabbit Alexa-Fluor-488 and donkey anti-goat Alexa-Fluor-555 (1:100, Jackson ImmunoResearch).

**cDNA library production.** Cells from the SNpc and VTA respectively were isolated using LCM (Leica AS-LMD) from stained frozen sections of control and Lmx1a/b cKo mice as described above. Microdissected cells were collected in lysis buffer, and total RNA isolation was carried out using an RNA isolation kit (Arcturus PicoPure, Applied Biosystems) according to the manufacturer's instructions. RNA was reverse-transcribed using Superscript III (Life Technologies) and the cDNA obtained was used for RT-qPCR.

**RNA sequencing.** RNA sequencing was performed on three biological replicates for the control and four biological replicates for Lmx1a/b cKO mice. For each sample, two technical replicates were performed. For each E15.5 embryo used, VTA and SNpc were dissected from eight antero–posterior levels across the entire DA domain as revealed by TH immnostaining. Following RNA extraction, RNA-seq libraries were constructed using the Illumina TruSeq Stranded RNA protocol with oligo dT pulldown and sequenced on Illumina HiSeq2500 by 150-bp paired-end sequencing. One sample was excluded before the sequencing because the amount of RNA was too low to measure the RNA quality. Reads were aligned using STAR aligner (version 2.4.2a) to the mouse genome assembly GRCm38. Raw read counts per gene were obtained using featureCount in the R package Rsubread with the GENCODE annotation vM8. The gene differential expression analysis was done using DeSeq2[50]. We only considered genes with >10 reads across all of the samples. A total of 225 genes with adjusted $p$ value <0.2 were reported as being significantly differentially expressed. Data are available at Array Express under accession number E-MTAB-5986.

**RT-qPCR.** Quantitative RT-PCR was performed using a cDNA library obtained from the LCM experiment. Analysis of expression levels of mRNAs was achieved with Platinum SYBR Green Super Mix (Invitrogen) and performed in triplicate using the LightCycler 480 (Roche Diagnostics). Primers for amplification were designed in the 3′ region of each gene using the online Primer3 tool (http://www.bioinfo.ut.ee/primer3/). The primer sequences are available upon request. Amplifications were performed in 20 μl containing 0.5 μM of each primer, 0.5 μl SYBR Green (Invitrogen), and 2 μl of 50-fold diluted cDNA, with 40 cycles at 94 °C for 15 s, 60 °C for 1 min, 72 °C for 30 s, and 79 °C for 5 s. Analysis of real-time quantitative RT-PCR triplicate reactions was performed with the LCS 480 software (Roche Applied Science; version 1.5.0.39); outliers were removed according to the method described by ref.[51]. The "Advanced Relative Quantification" mode of the LCS 480 software was used to estimate relative gene expression; the 2−ΔCT formula was used to calculate relative quantifications. Values were normalized using the amount of target gene for at least two or three reference genes (GAPDH, TBP, and RPL13). To confirm homogeneous product formation, a melting curve analysis was performed. Data are represented as the mean ± SEM of the fold change normalized against reference genes.

**Brain injections.** The anterograde virus AAV2-mCBA-FLEx-EGFP-WPRE (9 e12 genome copy per ml; from Plateformed'OutilsMoléculaires, IUSMQ) was injected into the SNpc of Dat$^{Cre/+}$ Lmx1a/b$^{f/f}$ or Dat$^{Cre/+}$ mice at P10. Mice were anesthetized using isoflurane and immobilized in a stereotaxic apparatus (Stoelting, Wood Dale, IL, USA). Then, 200 nl of this virus was injected at the following coordinates: +1 mm anteroposterior (AP) from lambda; 1.1 mm mediolateral (ML); −3.2 mm dorsoventral (DV, taken from the surface of the brain)[52]. Mice were killed 17 days after surgery for immunohistochemistry analysis. Injection of AAV2-Retro-GFP (2 e13 genome copy per ml; from Plateforme d'Outils Moléculaires, IUSMQ) was performed in the striatum of the Dat$^{Cre/+}$ Lmx1a/b$^{f/f}$ or Dat$^{+/+}$ Lmx1a/b$^{f/f}$ mice, Sema7a KO, or WT at P10. Two hundred nanoliters was injected in both hemispheres at the following coordinates for the ventral striatum: bregma: +0.7 mm AP; ±1 mm ML; −3.5 mm DV; and for the dorsal striatum: bregma: +0.7 mm AP; 1.6 mm ML; −2 mm DV[52]. Mice were killed 17 days after surgery for analysis. For the Fluorogold injections in the ventral striatum, we used the same coordinates, but we injected 1 μl of Fluorogold (2% in 0.9% sodium chloride, Millipore) in both hemispheres and mice were killed 3 days after.

**Stereological neuron counting.** Stereological methods (optical fractionator method, StereoInvestigator, MBF Bioscience)[53–55] were used to compare the number of TH-positive neurons within the ventral midbrain of control and mutant mice at P1 and P15. Contours were traced at ×5 magnification to define the area of interest, and counting was performed at ×40 magnification at one in two section intervals. Coefficients of error (Gundersen, $m = 1$) were <0.08.

**Western blot**. Striata from control $Pitx3^{GFP/+}$ mice brains at P1 were dissected by cutting a 1-mm coronal slice, followed by dissection of samples from dorsal and ventral striata. These samples were then snap-frozen. Sample lysis was performed in radioimmunoprecipitation buffer complemented with protease inhibitor and phosphatase inhibitor cocktails (Roche) (50 mM Tris-HCl, 150 mM NaCl, 1% NP-40, 0.5% sodium deoxycholate, 0.1% SDS, 2 mM EDTA, 50 mM NaF, pH 7.4). Quantification of protein content in the samples was carried out using a DC-protein assay (Bio-Rad). Fifteen micrograms of protein extracts were separated by SDS-polyacrylamide gel electrophoresis (12% SDS-PAGE Tris-glycine gels). Nitrocellulose membranes (Bio-Rad) were used for the transfer. Blots were immunostained overnight at 4 °C with primary antibodies. The primary antibodies —mouse anti-actin (1/10000; Millipore, MAB1501) and rabbit anti-Sema7a (1/500; Abcam, ab23578)—were diluted in blocking solution containing TBS and 0.01% Tween-20 (Sigma). Blots were washed three times (5 min each) with TBS and 0.01% Tween-20, and immune complexes were detected with species-appropriate secondary antibody conjugated to HRP: Goat anti-rabbit HRP (1/3000; CST 7074) and Goat anti-mouse HRP (1/5000; Life Technologies, G-21040). Membranes were covered with ECL for 5 min (Western Lightning Plus-ECL, PerkinElmer), and chemiluminescence was then documented by exposing the membranes to Pierce CL-Xposure film (Thermo Scientific). Films were scanned and analyzed using the ImageJ64 program.

**In vitro ventral midbrain cell cultures and explant cultures**. Primary ventral midbrain cell cultures were prepared as previously described[8]. In brief, embryonic ventral midbrains were dissected from E18.5 Pitx3-GFP embryos in chilled L15 containing 5% fetal bovine serum (FBS). After one DIV, cells were transfected using Lipofectamine 2000 (Life Technologies) with either the control plasmid pCMV-mCherry (Plateformed'OutilsMoléculaires, IUSMQ), or a plasmid allowing overexpression of Otx2 (gift from S.-L.A.), Lmx1a (pCAGGS-Lmx1a-IRES2-nucEGFP, Addgene), or Lmx1b (pCMV-Lmx1b-RFP, Applied Biological Materials Inc.). Cells were collected after three DIVs and processed for RT-qPCR.

Explants of embryonic ventral midbrain were dissected from E14.5 $Pitx3^{GFP/+}$embryos in chilled L15, 5% FBS. Explants were grown on 12-mm diameter glass coverslips coated with 7 μl of Matrigel™ (BD Biosciences, Mississauga, ON, Canada) in 24-well plates. Then, explants were each covered by 7 μl of Matrigel and cultured in 1 ml of neurobasal medium complemented with B27, PenStrep, Glutamax, sodium pyruvate, and FBS (86.8% neurobasal medium, 5% P/S, 2% B27, 0.2% Glutamax, 1% Na-Pyruvate, and 5% FBS) for 2 days at 37 °C, 5% $CO_2$. After two DIVs, the medium was replaced by the same complemented medium without FBS. Then, explants were exposed to Sema7a (0.5 μg/ml) for 2 h, and fixed for 30 min at 4 °C in fixative solution (3.5% PFA, 4% sucrose, in PBS 1×). The actin was stained by incubating the explants for 40 min at room temperature with a buffer solution containing 1% rhodamine-phalloidin (Life Technologies) and 1% BSA, while TH was immunostained by incubating overnight with rabbit anti-TH (Pel-Freez, 1:1000) in 5% NDS and 0.1% Triton X-100 in PBS. Axonal outgrowth analysis was performed on confocal images of the TH signal using the Neurite-J plug-in[56].

**Stripe assay**. The stripe assay was performed as previously described[57]. Alternating stripes (IgG or Sema7a 100 μg/ml, R&D system) were applied to glass coverslips. These stripes were then covered by laminin (20 μg/ml). Approximately three to four explants from $Pitx3^{GFP/+}$embryos or from Lmx1a/b cKO and control littermates were seeded in the middle and at the left and right extremities of each coverslip. These explants were then cultured for 3 days, and fixed with 3.5% PFA, 4% sucrose, in DPBS (1×) before immunostaining. For quantification, the number of neurites terminating on control vs. Sema7a stripes was counted for each explant.

**iDISCO**. Mice brains were dissected at P1 and then processed as described in the iDISCO protocol[20]. Briefly, after tissue collection and fixation, a pretreatment with methanol was performed before immunostaining. Then, permeabilization and blocking of the tissue were performed before incubation in primary antibodies at the following concentration: rabbit anti-TH (Pel-Freez, 1:250) for 7 days. After 1 day of washing in buffer PTwH (100 ml PBS 10×, 2 ml Tween-20, 1 ml of 10 mg/ml heparin stock solution), brains were incubated in the secondary antibody Alexa Fluor 594 donkey anti-rabbit (Life Technologies, 1:400) for 6 days. After washing, brains were transferred in a clearing solution for further imaging.

**Light-sheet imaging**. Whole hemisphere three-dimensional (3D) images were obtained using a two-photon light-sheet microscope. An excitation light-sheet was generated by scanning a long, thin near-infrared (800 nm) Bessel beam into the sample through a long working distance objective (Olympus, XLFLUOR4X/340, NA 0.28, WD 29.5 mm). The length and thickness of the beam were decoupled, thereby achieving a large field of illumination (2.5 mm × 2.5 mm) without compromising the axial resolution (2 μm). To produce a light-sheet with sufficient energy, a femtosecond titanium-sapphire laser source (Coherent Mira 900) was used with an amplifier (Coherent RegA 9000). The emitted two-photon fluorescence was detected by a scientific CMOS camera (Hamamatsu Orca-Flash4.0 V2) through a long working distance multi-immersion objective (Olympus XLPLN10XSVMP, NA 0.6, WD 8 mm) placed perpendicular to the scanning plane, achieving an effective field of view of 1.3 mm by 1.3 mm. The whole-brain images were obtained by moving the sample using a motorized stage (Sutter MPC-385), thus generating 3D images. These images were stitched together using a custom Matlab script to form a 3D image of the entire sample.

**Electrophysiological recordings**. Sagittal whole-brain slices (300 μm thick) were prepared from $Dat^{Cre/+}Lmx1a/b^{f/f}$cKO mice and control littermates (P9–14). Animals were lightly anesthetized with isoflurane and brains were dissected and sliced in ice-cold (0–4 °C) solution containing (in mM): 250 sucrose, 2 KCl, 1.25 $NaH_2PO_4$, 26 $NaHCO_3$, 7 $MgSO_4$, 0.5 $CaCl_2$, and 10 glucose, saturated with 95% $O_2$ and 5% $CO_2$, 340–350 mOsm. Slices were immediately transferred to a heated (34 °C) oxygenated solution containing (in mM): 126 NaCl, 2.5 KCl, 1.4 $NaH_2PO_4$, 26 $NaHCO_3$, 2.4 $MgCl_2$, 1.2 $CaCl_2$, and 11 glucose (60 min, after which they were kept at room temperature until use). During recordings, slices were continuously perfused at 2 ml per min with a standard artificial cerebrospinal fluid (126 mM NaCl; 2.5 mM KCl; 1.4 mM $NaH_2PO_4$; 25 mM $NaHCO_3$; 1.2 mM $MgCl_2$; 2.4 mM $CaCl_2$; and 11 mM D-glucose; osmolarity adjusted to 295–305 mOsm) saturated with 95% $O_2$ and 5% $CO_2$ at near-physiological temperature (30–32 °C). Whole-cell voltage-clamp recordings were obtained from visually identified cells. For recordings, 3.5–5 MΩ borosilicate glass pipettes were filled with a 115 mM CsMeSO3, 20 mM CsCl, 10 mM diNa-phosphocreatine, 10 mM HEPES, 2.5 mM $MgCl_2$, 0.6 mM EGTA, 4 mM ATP-Mg, and 0.4 mM GTP-Na (pH 7.25; osmolarity adjusted to 275–285 mOsm). Data acquisition (filtered at 2–3 kHz; digitized at 10 kHz) was performed using a Multiclamp 700B amplifier and Clampex 10.6 software (Molecular Devices, Union City, CA, USA). Data were analyzed using Clampfit 10.6 (Molecular Devices). Spontaneous and miniature events were analyzed using the search event algorithm in Clampfit 10.6 and fitted with a double exponential.

**Statistical analysis**. Statistical analyses were performed using GraphPad Prism 4.0 (GraphPadSoftware, La Jolla, CA, USA) software. The differences between two groups were determined by Student's $t$ test or Mann–Whitney $U$ test when variance was different between groups. Explant and mDA cell intersection profiles were analyzed by two-way analysis of variance, and the Sidak test was used for post hoc comparisons. All data are represented as means ± SEM, and significance is defined as *$p < 0.05$, **$p < 0.01$, or ***$p < 0.001$.

**Microscopes**. All immunofluorescence images were acquired using a Zeiss LSM5 Pascal confocal microscope ora Zeiss LSM700 confocal microscope, then processed using Zen software and Adobe Photoshop CS4. Brightfield pictures were acquired using a Leica DMRB equipped with a digital camera. Tiled brightfield images were also acquired using an Olympus BX51 equipped with a motorized stage and stitched with Surveyor Software for Real-Time Mosaic Imaging (Objective Imaging, Cambridge, UK).

**Data availability**. All data generated and analyzed during the current study are available from the corresponding author on request. RNA sequencing data are available at Array Express under accession number E-MTAB-5986.

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

## Acknowledgements

This work was mainly supported by a grant from the Natural Sciences and Engineering Research Council of Canada (NSERC: 418391-2012) and by Canadian Institutes of Health Research Grants MOP 311120 to M.L. A.C. received a scholarship from the Fondation de l'institut universitaire en santé mentale de Québec, and from the Centre thématique de recherche en neurosciences (CTRN). M.L. is a FRSQ Chercheur-Boursier. R.J.P. received funding from Stichting ParkinsonFonds and the Netherlands Organization for Scientific Research (ALW-VICI). S.-L.A. is supported by the Francis Crick Institute, which receives its core funding from Cancer Research UK (FC001089), the UK Medical Research Council (FC001089), and the Wellcome Trust (FC001089); and by Parkinson's UK research grant G0617. E.M. is funded by Parkinson's Disease UK (F1501). We thank Veronique Rioux for technical assistance. We thank Meng Li for sharing Pitx3-GFP mice and Thomas Perlmann for Lmx1a flox mice. We thank Dr Armen Saghatelyan for his useful comments and Dr Louis-Eric Trudeau for sharing Sema7a KO mice. We also thank the Plateforme d'Outils Moléculaires (https://www.neurophotonics.ca/fr/pom) for the production of the viral vectors used in this study.

## Author contributions

A.C. analyzed mutant mice, performed, and analyzed most of the histological preparations, in situ hybridization, immunostainings, explants and stripe assay, and retrograde and anterograde tracing experiments, and performed dissection and genotyping for all mice used in this study. A.C. and M.L. wrote the article. A.C. and H.D.-B. performed the LCM and made the cDNA library for LCM samples; H.D.-B. and E.M. performed and analyzed qRT-PCR experiments. G.B. performed and analyzed explant experiments. A.D. performed retrograde labeling experiments on Sema7a KO. C.S. performed and analyzed electrophysiological experiments. E.M., M.I.A., and M.L. performed and analyzed mRNA sequencing at E15.5. M.S.P. performed and analyzed western blot experiments, neuron counting at P15, and stripe assay on Lmx1a/b cKO embryos. A.C., G.B. and J.C. performed mDA primary cell culture experiments. C.A., D.C. and M.L. built the light-sheet microscope and C.A. performed imaging of transparent brains. S.L. and R.J.P.

performed Plxnc1 expression analysis on Sema7a KO. Funding acquisition, M.L.; S.-L.A., E.M. and M.L.. supervised the study. All authors contributed to manuscript preparation, read and approved the final manuscript.

## Additional information

**Competing interests:** The authors declare no competing financial interests.

