## [Peer Review File · Nature Communications]

Reviewers' expertise:

Reviewer #1: axon guidance;

Reviewer #2: development of midbrain DA neurons;

Reviewer #3: development of midbrain DA neurons;

Reviewer #4: neural development, axon guidance.

Reviewers' comments:

Reviewer #1 (Remarks to the Author):

In this manuscript by Chabrat et al., the authors investigate the roles of the transcription factors Lmx1a and Lmx1b in dopaminergic (DA) neurons pathfinding. They first show that Lmx1a and Lmx1b are both expressed in developing DA neurons, including the VTA and SN. To specifically explore the roles of these transcription factors in late steps of neuronal differentiation they perform conditional double mutants (dKO) using the DAT-cre line, which drives recombination from E14 onwards. In dKOs, TH-positive DA axons are lacking in the dorsalmost half of the striatum in P1 pups. In addition, in dKOs dorsal striatal neurons have a lower frequency in mEPSCs and miPSCs than in controls and ventral striatal neurons higher ones. Using anterograde viral injections and retrograde fluorogold labeling, the authors show that DA neurons project to reduced regions of the striatum and that more SN neurons project ventrally in the dKO. In search for the factors regulating axonal pathfinding downstream of Lmx1a and Lmx1b, they perform laser-capture and RNA-seq on DA neurons from controls and dKOs and identify Plexinc1 (plxnc1) as an interesting candidate gene. Plxnc1 is expressed only in the VTA and is upregulated in both SN and VTA in dKO. Using gain of function experiments in DA neurons primary cultures, the authors show that Lmx1a/b repress plxnc1 expression. The authors next examined through in vitro assays the effect of sema7a on VTA and SN neurons and show that full sema7a knock-out affects the targeting of plexinc1-expressing axons. Conversely, transgenic over expression of plexinc1 mimicks the phenotype observed in Lmx1a/b dKOs. Last but not least the authors show that the VTA-restricted expression of Otx2 expands to the SN in Lmx1a/b dKO and that Otx2 is also involved in regulating plxnc1 expression.

The topic of the manuscript, how DA neurons are specified and guided to their appropriate targets is of great interest. This manuscript has thus a high potential to impact the field.

The experiments performed are of high quality, and use distinct technological approaches to investigate specific points. The article is well-written and the main claim important.

However, in its present state there are a series of issues, loose ends and over-interpretations (see below) that the authors must address before the paper is suitable for publication.

Major points:

- The phenotypes of several of the mutant lines are NOT appropriately schematized on the figures or depicted in the text, which is highly confusing for the reader and leads to misinterpretations. For instance, the phenotype described in Lmx1a/b dKO in figure 1 consists of a reduction in the density of TH fibers in the dorsal part of the dorsal striatum (1d, 1i and 1j). However, the authors present in the drawing in fig1c an increased axonal density in the ventral striatum, which is not supported by their data. The authors should modify that figure as well as analyze the axonal innervation at another timepoint to establish whether it is a targeting defect or delay, and whether there is an excess of axons targeting ventrally.

- The main argument to state that Lmx1a/b impact on axonal targeting is the retrograde fluorogold

injection in the dorsal striatum versus accumbens (Figure 3g-j). This experiment has several caveats: i) fluorogold is extremely leaky, so it is difficult to ascertain the zone of injection; ii) there is no control for the injection sites; iii) the quantification (SN/VTA ratio) is not fully explained; iv) the injection are performed in the accumbens versus striatum, a region much more ventral than the one examined in the rest of the article. It is mandatory that the authors show convincing data on this specific point and clarify which region of the striatum they are examining (the accumbens, the dorsal part of the dorsal striatum, the ventral part of the dorsal striatum etc...).

- The expression pattern of Sema7a in figure 5a should be presented at birth or before (instead of P4), since the axonal phenotypes are already present at this stage (fig1). Overall, the splitting into two subregions of the striatum deserves a more specific justification. Is it the delineation of VTA versus SN projections? The authors should at least explain their arbitrary delineation, because it does not correspond to the estimated limit between the accumbens and striatum. In addition, they should change the title of figure 5, because Sema7a is expressed, at lower levels (see panel 5b), but is expressed in the ventral striatum.

- In figure 7a-h, the authors examine the extension of PlxnC1-positive axons in controls and Sema7a full KO. An important control would be to check that PlxnC1 expression is indeed restricted to the VTA in these mutants.

- In figure 7j-o, the authors examine the impact of transgenic overexpression of Plexinc1 in DA neurons. I have a hard time understanding why the values of TH intensity ratio D/V are so different in controls presented in fig 6o than in the controls presented in fig2j (more than twice). It is important that the authors explain how the data was quantified, why controls vary so much, especially when the differences in axonal targeting are not that striking.

- The results presented in figure 7 raise a lot of questions. The authors show that the VTA-specific transcription factor Otx2 is expressed in the SN of Lmx1a/b dKO and can drive the expression of Plexinc1 in these neurons. It is thus tempting to speculate that SN neurons are transformed into VTA neurons in Lmx1a/b dKO and that the differential guidance of the two populations are regulated by Plexinc1. The way the results are currently presented is not fully supported by the results and highly over interpreted. It would be important to determine whether Otx2 is mandatory for the upregulation of plexinc1 in Lmx1a/b dKO.

On a more general note, the authors put forward that the phenotypes observed are axonal targeting defects. However, they usually only show only one time point (it could be a change in growth, branching or synaptogenesis (leading to a reduced density of terminals in the zone of interest)). They should probably down tone this statement or provide stronger experimental evidence.

Minor points:

- DAT-cre recombination is extremely sparse at E13.5- the authors might want to either check that issue or modify the text for E14.
- The legend of Figure 2n is missing.
- Figure S4 is not convincing and should be properly quantified.

Reviewer #2 (Remarks to the Author):

In this manuscript Chabrat et al. have studied the function of the transcription factors Lmx1a and Lmx1b (Lmx1a/b) during dopamine (DA) neuron maturation by studying mice in which Lmx1a/b have been ablated after crossing with DAT-Cre mice. The results indicate a rather unexpected role in

dopaminergic circuit formation. Although *Lmx1a/b* are expressed in all DA neurons and are essential in all DA neural progenitors, the results in this manuscript show that *Lmx1a/b* ablation leads to absence of DA neuron terminals specifically in the dorsal striatum. A number of experimental approaches indicate that DA neurons in the SNpc are unable to innervate the dorsal striatum in KO animals due to a failure to repress *Plxnc1* in the SNpc in the absence of *Lmx1a/b* activity. When *Plxnc1* becomes expressed in SNpc in the mutant animals, neuron terminals are repelled from the dorsal striatum as consequence of interactions with the ligand semaphoring 7a (*Sema7a*).

Overall this is an interesting and extensive study that goes a long way to explain an important aspect of how circuit formation of VTA and SNpc DA neurons is regulated. Understanding DA neuron circuit development is an important and poorly understood feature of embryogenesis. As the authors point out, the results have potential clinical significance, e.g. in efforts to develop cell replacement therapy for Parkinson's disease. However, before publication a number of issues need to be addressed to improve the manuscript and solidify the conclusions.

Major comments

1. The computational methods used to analyze the RNAseq experiment need to be explained. As it stands it is not possible to evaluate the quality of these data (see also minor point 6 below). The number of samples from each animal that were sequenced needs to be indicated. From what levels of the ventral midbrain were samples dissected? How many animals were used? It is also important to show data indicating the quality of the RNAseq data by showing Spearman correlation or another statistical method to indicate how similar samples are to each other. A volcano plot or scatter diagram should be displayed to visualize the correlation between KO and controls. The statistical method(s) used to identify differentially expressed genes should be indicated and explained. It is also important to include tables of all differentially expressed genes, both the 517 that are higher in the mutant, and those that are expressed at a lower level. It is not acceptable to only show a few selected genes as a Supplementary Table 1. All of the requested data can easily be added as supplementary information, and they will be absolutely essential for readers of a published paper. The original data should also be deposited to a public data base at the time when the paper is published.

2. In Figure 7, the authors have quantified the dopaminergic axon terminals in the striatum of *Sema7a* KO by *Plxnc1* staining. Why was TH staining not used instead, to make the results more comparable with those in Figure 2?

Moreover, although the authors state that *Plxnc1* is restricted to VTA mDA neurons (line 202), it is actually widely expressed elsewhere in the brain (Allen Brain Atlas). Therefore, in *Sema7a* KO mice, *Plxnc1*-expressing neurons from other areas might be misguided to the dorsal striatum. The authors should investigate this possibility by performing fluorogold injections, similar to those shown in Figure 3, to *Sema7a* KO animals.

Minor comments

1. In Figure 1a, the scale bar in the upper high magnification insert appears to be 10 μ m than 100 μ m as stated in the figure legend.

2. The authors describe their primary cell cultures from *Pitx3-EGFP/+* animals as primary mDA neuron cultures. However, as they did not sort for GFP+ cells, the actual mDA neurons represent only a subset of the neurons in culture – and the observed gene expression changes represent the outcome of over expression in various types of neurons (and possibly glia), not only in mDA neurons. This should be pointed out. It is also more correct to refer to the cells as “ventral midbrain” primary neuron cultures.

3. In Figure 2d and 2n, it appears that there are clearly more axonal terminals ending in the ventral striatum in P1 mutants – as might be expected if SNpc neurons are misguided towards this region. However, in Figure 2i and j, the quantification results do not show this. The authors could comment on this.
4. The figure legend for Figure 2n is absent. The iDisco images appear a bit dark and unclear and might benefit from drawing a light outline to visualize the shape of the striatum.
5. In line 131, the control mice should be *DatCre/+* instead of *Dat+/+*, according to the label in Figure 3b.
6. The LCM experiment should be described in more detail. How many pieces of SNpc and VTA were collected from each embryo, from which antero-posterior levels, and from how many embryos?
7. In Figure 6h and j, the stripes are not clearly visible. The text for IgG and Sema7a should be bigger and the position of the stripes indicated. In line 193 in the text, it should probably read Fig. 6f,g instead of Fig. 6g,h.
8. In many parts of the text, the authors describe the results in an odd order. For example, in Figure 6, the authors start from 6i-k, moving on to 6e,f, then to 6a-d, and going back to 6g,h. This makes the text difficult to follow. The figures and/or text should be reorganized so that the results are presented and described in logical order.
9. Throughout the paper, the authors should clearly state in each figure legend the number of embryos used for each quantification experiment.
10. In earlier study (Laguna et al., Nat. Neuroscience 2015), the mice with the same genotype (*Dat-Cre/+; Lmx1aflox/flox; Lmx1bflox/flox*) do not display similar drastic loss of dorsal striatal innervation. The authors should discuss this – could this be due to, for example, a different genetic background of the mice? Have different strains with different floxed alleles been used? The genetic background of the mouse strains in this study has not been mentioned in the Methods.

Reviewer #3 (Remarks to the Author):

The manuscript by Chabrat et al reports an interesting study showing the role of Lmx1a and Lmx1b in directing axon innervation of mdDA neurons to the forebrain. In this manuscript the authors show a huge number of logic experiments. They first analyze mice conditionally lacking Lmx1a and Lmx1b in post-mitotic *Dat-Cre+* neurons. These mutants exhibit loss of mdDA axons in the dorsal and posterior striatal region; this finding is also supported by a number of additional experiments including anterograde and retrograde tracing experiments. Then, by RNA-seq experiments, the authors identify the axon guidance receptor *Plxnc1* as a potential transcriptional target of Lmx 1a and 1b factors. Importantly, they show that *Plxnc1* can be down-regulated by Lmx factors and that in *Lmx1a/1b* double KO mice *Plxnc1* is activated also in SNpc while normally it is expressed only in VTA. Since *Plxnc1* can be bound by Sema 7a, the authors performed a further series of experiments. They first show that i) the level of Sema 7a is higher in the dorsal posterior striatum compared to the ventral striatum; ii) exposure of VTA and SNpc explants to Sema 7a impairs the length and complexity of VTA axons while these parameters are not affected in SNpc explants; and iii) Sema 7a KO mice exhibit a more dorsal distribution of *Plxnc1+* axons. These data are further reinforced by *in vivo* *Plxnc1* over expression in SNpc and VTA neurons.

Finally, the authors show that Otx2, whose expression is restricted to VTA, is involved in maintenance/activation of Plxnc1 by antagonizing the repressive role of Lmx factors. Indeed, Otx2 may induce Plxnc1 expression but it is not repressed by Lmx 1a and 1b factors. Remarkably, Otx2 is activated in SNpc neurons of Lmx double KO mice, which, in turn, supports Otx2-mediated activation of Plxnc1 in SNpc.

I think that the data reported in this manuscript are of high interest, elucidate a specific mechanism controlled by Lmx1a and 1b genes, and are of particular relevance to understand how the nigrostriatal and mesolimbic pathways are specified. In this context, this manuscript should result of high impact and be of general interest. Nevertheless, and although very dense of experiments, I would like to ask 3 additional experiments which should further improve its quality:

1) The experiments on VTA and SNpc explants to assay the effect of exposure to Sema 7a, should be performed also in explants from Lmx double KO.

2) Is the Otx2 expression in VTA and SNpc neurons of Lmx double KO mice co-localized with Plxnc1?

3) Is the Plxnc1 expression maintained in VTA neurons of mice lacking Otx2? Indeed, based on the data here reported, it should be expected a phenotype mirroring that observed in Lmx double KO.

In my opinion, with these improvements this manuscript should be published in Nature Communication.

Reviewer #4 (Remarks to the Author):

The authors provide convincing data that Lmx1a/1b together with Otx2 determine levels of the axon guidance receptor Plxnc1 in the VTA dopamine neurons, and that these factors together with an additional unknown molecule control the levels of Plxnc1 in the SN as well. These combinations of factors result in high levels of Plxnc1 expression in the VTA and no expression in the SN. In the striatum, the axonal target area of mDA neurons, there is high expression of the Plxnc1 ligand Sema7a, restricted to the dorsal anterior striatum. They provide evidence that Sema7a functions as an inhibitory cue for VTA but not SN axons in their assay. They propose that that directs segregation of VTA axons and SN axons: VTA axons with high Plxnc1 avoid the Sema7a rich region, whereas SN axons innervate the Sema7a rich region.

The authors did a comprehensive set of studies including KO and overexpression of Lmx1a/b, KO of Sema7a, and examination of the response of neurons and explants to Sema 7a in vitro.

The data are clear and convincing, and on the whole the paper is well written. It provides an additional set of cues that DA axons use to determine which region of the striatum they will innervate. The striking similarity of the phenotype in Lmx1a/1b KO mice to the previously reported Netrin1 KO mice is interesting and suggests that both sets of molecules play important roles.

I believe the paper needs a few minor textual changes indicated below. I also suggest an optional experiment that could enhance the interpretation.

1. Otx 2 appears to be a major player in the ultimate regulation of Plxnc1 expressed by VTA and SN neurons. Although well addressed in the discussion, it would be preferable also to mention Otx2 and its role in the abstract and introduction.

2. Since the difference in outgrowth of SN and VTA neurons in the presence of Sema7a in vitro did not differ very much using single neurons and Sholl analysis for quantification, it would be nice to see SN explants grown on stripes of Sema7a. It is expected that SN axons would ignore the boundary and grow equally well on both control and Sema 7a stripes. If not, the interpretation may be different.

3. In wild-type animals, if Sema7a is preventing the VTA axons from innervating the dorsal striatum, what is preventing the SN axons from innervating the ventral striatum? It may be that as shown in ref 36, high Netrin-1 levels in ventral striatum prevents these axons from growing ventrally. Rather than wait and mention this study in the discussion it would seem more appropriate to cite in the introduction and discuss in the discussion (including discussing the Netrin1 KO phenotype, which currently is not discussed).

4. The authors should cite other papers in this paragraph of the Introduction:
"Very little is known about the molecules regulating axonal targeting of mDA neurons, and how the different mDA neuronal populations establish their specific connections is unknown."

With proper attention to these minor points, the paper is appropriate for Nature Communications.

Reviewer 1

Major point 1:

- The phenotypes of several of the mutant lines are NOT appropriately schematized on the figures or depicted in the text, which is highly confusing for the reader and leads to misinterpretations. For instance, the phenotype described in Lmx1a/b dKO in figure 1 consists of a reduction in the density of TH fibers in the dorsal part of the dorsal striatum (1d, 1i and 1j). However, the authors present in the drawing in fig1c an increased axonal density in the ventral striatum, which is not supported by their data. The authors should modify that figure as well as analyze the axonal innervation at another timepoint to establish whether it is a targeting defect or delay, and whether there is an excess of axons targeting ventrally.

We agree with the reviewer and all schematics have been corrected.

In addition, we now provide quantification of TH axon density in the striatum at postnatal day 15 (**Supplementary Fig. 2**). We confirmed that DA axonal density is lower in the dorsal striatum of Lmx1a/b cKO, and as in P1, we did not observe an increase of DA axons density in the ventral striatum at P15.

- The main argument to state that Lmx1a/b impact on axonal targeting is the retrograde fluorogold injection in the dorsal striatum versus accumbens (Figure 3g-j). This experiment has several caveats: i) fluorogold is extremely leaky, so it is difficult to ascertain the zone of injection; ii) there is no control for the injection sites; iii) the quantification (SN/VTA ratio) is not fully explained; iv) the injection are performed in the accumbens versus striatum, a region much more ventral than the one examined in the rest of the article. It is mandatory that the authors show convincing data on this specific point and clarify which region of the striatum they are examining (the accumbens, the dorsal part of the dorsal striatum, the ventral part of the dorsal striatum etc...).

We agree with the reviewer that it is very difficult to visualise the injection site with Fluorogold. In addition to our anterograde AAV-GFP Flex experiments confirming the targeting defect, we performed experiments using the recently developed retrograde adeno-associate viral vector variant allowing efficient retrograde labeling (Tervo et al., et al., Neuron 2016, Ref. 24).

i-ii) With this retrograde AAV-Retro-GFP we could visualise the injection site (Fig. 3h and 7l) and measure DA neurons retrogradely labelled in the VTA and SNpc (Fig. 3k-n and 7m-p).

iii) We now provide a more detailed explanation of the quantification (see method section).

iv) We now describe the targeted regions. For Lmx1a/b cKO mice analysis, AAV-Retro-GFP has been injected in the ventro-medial striatum whereas for the analysis of Sem7a KO we injected the AAV-Retro-GFP in the dorsal striatal region.

- The expression pattern of Sem7a in figure 5a should be presented at birth or before (instead of P4), since the axonal phenotypes are already present at this stage (fig1). Overall, the splitting into two subregions of the striatum deserves a more specific justification. Is it the delineation of VTA versus SN projections? The authors should at least explain their arbitrary delineation, because it does not correspond to the estimated limit between the accumbens and striatum. In addition, they should change the title of figure 5, because Sem7a is expressed, at lower levels (see panel 5b), but is expressed in the ventral striatum.

We agree with the reviewer and we now present the expression pattern at embryonic day 18.5 and at postnatal day 4 (Fig. 5a-b).

Regarding the striatal subregions, we based our choice on axon tracing experiments (Ref. 2, 3, 5, 48). Although there is no clear boundaries between SNpc and VTA target regions in the striatum, the dorsal half of the striatum receive mostly projections from SNpc whereas the ventral half receives mostly projections from VTA. We added a justification in the method section.

Regarding the Figure 5, we agree with the reviewer and we changed the title of the figure.

- In figure 7a-h, the authors examine the extension of PlxnC1-positive axons in controls and Sema7a full KO. An important control would be to check that PlxnC1 expression is indeed restricted to the VTA in these mutants.

We now provide an analysis of Plxnc1 in VTA and SNpc of controls and Sema7a mutants (see supplementary Fig. 9). We do not observe difference in the number of DA neurons containing Plxnc1 in Sema7a KO mice.

- In figure 7j-o, the authors examine the impact of transgenic overexpression of Plexinc1 in DA neurons. I have a hard time understanding why the values of TH intensity ratio D/V are so different in controls presented in fig 6o than in the controls presented in fig2j (more than twice). It is important that the authors explain how the data was quantified, why controls vary so much, especially when the differences in axonal targeting are not that striking.

We now provide a more detailed explanation of our axonal density measurements (see method section). The measurements were made using Li-Cor imager that measures the infrared fluorescent signal for TH that we divided by the fluorescent signal for actin. This has been used in many publications (Ref. 46, 47) and produces a non-bias measurement of signal intensities.

It is hard to compare these measurements across different set of experiments for many reasons. First, It is possible that part of the variation in D/V ratio between controls are coming from the their different genetic backgrounds. The control animals used in Figure 7 are wild type animal with the same genetic background than the Sema7a KO and Plxnc1 OE mice. In Figure 2, the control mice used are $Dat^{+/+}$ $Lmx1a/b^{F/F}$. We also used matched sections between controls and mutants for 3 different AP levels. Between set of experiments, the anterior, central and posterior levels might have been slightly different. To avoid confusion we provided measurement as percent of control and we specified which control we used in each experiments.

- The results presented in figure 8 raise a lot of questions. The authors show that the VTA-specific transcription factor Otx2 is expressed in the SN of $Lmx1a/b$ dKO and can drive the expression of Plexinc1 in these neurons. It is thus tempting to speculate that SN neurons are transformed into VTA neurons in $Lmx1a/b$ dKO and that the differential guidance of the two populations are regulated by Plexinc1. The way the results are currently presented is not fully supported by the results and highly over interpreted. It would be important to determine whether Otx2 is mandatory for the upregulation of plexinc1 in $Lmx1a/b$ dKO.

We agree with the reviewer that it is important to determine whether Otx2 is required for the upregulation of Plxnc1 in $Lmx1a/b$ cKO. We thus used laser capture microdissection to isolate neurons that ectopically contain Otx2 in SNpc of $Lmx1a/b$ cKO and we measured the expression of Plxnc1 by RT-qPCR. We also measured Plxnc1 expression from the Otx2 positive DA neurons in VTA of both controls and $Lmx1a/b$ cKO as well as in TH neurons of SNpc from controls. We found that Otx2+ DA neurons in VTA from both controls and $Lmx1a/b$ cKO express Plxnc1 but most importantly, Otx2+ DA neurons from SNpc of $Lmx1a/b$ cKO embryos also express Plxnc1. Inversely, we could not detect Plxnc1 expression in SNpc of control embryos.

These data indicate that ectopically expressing Otx2 neurons in SNpc of Lmx1a/b cKO mice also express Plxnc1.

On a more general note, the authors put forward that the phenotypes observed are axonal targeting defects. However, they usually only show only one time point (it could be a change in growth, branching or synaptogenesis (leading to a reduced density of terminals in the zone of interest)). They should probably down tone this statement or provide stronger experimental evidence.

We agree with the reviewer comment and we are now providing analysis of TH immunolabeling at P15 (Supplemental Fig. 2) as well as retrograde labeling experiments in later time points for both Lmx1a/b cKO and Sema7a KO mice (Figs 3 and 7). Although our in vitro stripes assay data clearly show that the interaction of Sema7a/PlexinC1 regulates axon guidance of mDA neurons, we could not rule out the possibility that part of what we observed in vivo could come from a reduced axonal branching so we also try to down tone this statement in the text.

Minor points:

- DAT-cre recombination is extremely sparse at E13.5- the authors might want to either check that issue or modify the text for E14.

We agree and made the change in the text.

- The legend of Figure 2n is missing.

We added the legend for Fig. 2n, which is now Fig. 2j.

- Figure S4 is not convincing and should be properly quantified.

As DA cortical innervation is established mostly after birth, we decided to replace Figure S4 and we now provide a more convincing representation of extrastriatal targets as well as a quantification of TH density in each analysed regions at P15.

Reviewer #2 (Remarks to the Author):

Major comments

1. The computational methods used to analyze the RNAseq experiment need to be explained. As it stands it is not possible to evaluate the quality of these data (see also minor point 6 below). The number of samples from each animal that were sequenced needs to be indicated. From what levels of the ventral midbrain were samples dissected? How many animals were used? It is also important to show data indicating the quality of the RNAseq data by showing Spearman correlation or another statistical method to indicate how similar samples are to each other. A volcano plot or scatter diagram should be displayed to visualize the correlation between KO and controls. The statistical method(s) used to identify differentially expressed genes should be indicated and explained. It is also important to include tables of all differentially expressed genes, both the 517 that are higher in the mutant, and those that are expressed at a lower level. It is not acceptable to only show a few selected genes as a Supplementary Table 1. All of the requested data can easily be added as supplementary information, and they will be absolutely essential for readers of a published paper. The original data should also be deposited to a public data base at the time when the paper is published.

The number of samples from each animal and the number of animal used in the analysis have been added in the method section. The method used to identify differently expressed genes has also been added in the method section.

Initially, we identified Plxnc1 in an RNA seq (published last year) and we used only one control and one mutant. To provide a more solid representation of gene expression changes in Lmx1a/b

cKO we performed a new mRNA sequencing experiment using 4 Lmx1a/b cKO and 3 control embryos. As requested, we now provide a PCA plot, clustered heatmap, and an MA plot indicating the good quality of our data for this analysis. In a supplementary table we now provide the 225 genes that are differently expressed in Lmx1a/b cKO. As suggested, we will also deposit these data to a public database when the paper is published.

2. In Figure 7, the authors have quantified the dopaminergic axon terminals in the striatum of Sema7a KO by Plxnc1 staining. Why was TH staining not used instead, to make the results more comparable with those in Figure 2? Moreover, although the authors state that Plxnc1 is restricted to VTA mDA neurons (line 202), it is actually widely expressed elsewhere in the brain (Allen Brain Atlas). Therefore, in Sema7a KO mice, Plxnc1-expressing neurons from other areas might be misguided to the dorsal striatum. The authors should investigate this possibility by performing fluorogold injections, similar to those shown in Figure 3, to Sema7a KO animals.

We agree and we provide the analysis of TH staining and added the Plxnc1 analysis in supplemental.

As suggested, we also performed experiments where we injected AAV-GFP retrograde in the dorsal striatum. Quantification of retrogradely labeled neurons from the dorsal striatum revealed that a larger proportion of neurons containing the GFP were found in the VTA in the Sema7a KO than in the controls. These data are in agreement with the immunostaining analysis presented.

Minor comments

1. In Figure 1a, the scale bar in the upper high magnification insert appears to be 10 um than 100 um as stated in the figure legend.

The scale bar has been verified and corrected.

2. The authors describe their primary cell cultures from Pitx3-EGFP/+ animals as primary mDA neuron cultures. However, as they did not sort for GFP+ cells, the actual mDA neurons represent only a subset of the neurons in culture – and the observed gene expression changes represent the outcome of over expression in various types of neurons (and possibly glia), not only in mDA neurons. This should be pointed out. It is also more correct to refer to the cells as “ventral midbrain” primary neuron cultures.

The use of the Pitx3-GFP line allows a very precise dissection of the ventral midbrain DA domain and we obtain a very high concentration of DA neurons. However, we agree with the reviewer and we also obtain other cell types. We now refer in the text as ventral midbrain primary cell culture.

3. In Figure 2d and 2n, it appears that there are clearly more axonal terminals ending in the ventral striatum in P1 mutants – as might be expected if SNpc neurons are misguided towards this region. However, in Figure 2i and j, the quantification results do not show this. The authors could comment on this.

The reviewer is right that it seems that a small increase of TH terminals seems present in the ventral striatum of Lmx1a/b cKO mice. However, the quantification did not reach statistical significance.

4. The figure legend for Figure 2n is absent. The iDisco images appear a bit dark and unclear and might benefit from drawing a light outline to visualize the shape of the striatum.

We added the legend for Fig. 2n, which is now Fig. 2j. As suggested, we added a drawing to outline the shape of the striatum and added orientation arrows.

5. In line 131, the control mice should be *DatCre/+* instead of *Dat+/+*, according to the label in Figure 3b.

We made the correction in the text.

6. The LCM experiment should be described in more detail. How many pieces of SNpc and VTA were collected from each embryo, from which antero-posterior levels, and from how many embryos?

We added details on LCM and RNA seq in the method section.

7. In Figure 6h and j, the stripes are not clearly visible. The text for IgG and Sema7a should be bigger and the position of the stripes indicated. In line 193 in the text, it should probably read Fig. 6f,g instead of Fig. 6g,h.

We added dotted line between stripes and increased the text for IgG and Sema7a.
We also adjusted the proper lettering to each figures.

8. In many parts of the text, the authors describe the results in an odd order. For example, in Figure 6, the authors start from 6i-k, moving on to 6e,f, then to 6a-d, and going back to 6g,h. This makes the text difficult to follow. The figures and/or text should be reorganized so that the results are presented and described in logical order.

We made the corrections and now present and describe the results in logical order.

9. Throughout the paper, the authors should clearly state in each figure legend the number of embryos used for each quantification experiment.

We added the number of animals used in each quantification experiments.

10. In earlier study (Laguna et al., Nat. Neuroscience 2015), the mice with the same genotype (*Dat-Cre/+; Lmx1aflox/flox; Lmx1bflox/flox*) do not display similar drastic loss of dorsal striatal innervation. The authors should discuss this – could this be due to, for example, a different genetic background of the mice? Have different strains with different floxed alleles been used? The genetic background of the mouse strains in this study has not been mentioned in the Methods.

We used the same *Lmx1a* and *Lmx1b* flox mice than those used in Laguna et al 2015. However, we used a different *Dat-cre* and our mice were kept in a mixt genetic background. In the study by Laguna et al, they kept their mice in C57Bl6 genetic background. The different mice background might indeed influence the phenotype. It is however hard to compare the two studies since they do not show any analysis before 2 months of age, where neurodegeneration already occurs in these mice. They do observed an overall reduction in DA axonal density in striatum of 2 months old *Lmx1a/b* cKO mice but this can be due to neuronal death that occurs in these mice from 3 weeks after birth. We thus specified the genetic background of the mice used in our study.

Reviewer #3 (Remarks to the Author):

I think that the data reported in this manuscript are of high interest, elucidate a specific mechanism controlled by Lmx1a and 1b genes, and are of particular relevance to understand how the nigrostriatal and mesolimbic pathways are specified. In this context, this manuscript should result of high impact and be of general interest. Nevertheless, and although very dense of experiments, I would like to ask 3 additional experiments which should further improve its quality:

1) The experiments on VTA and SNpc explants to assay the effect of exposure to Sema 7a, should be performed also in explants from Lmx double KO.

We agree with the reviewer and we performed stripes assay with explants from Lmx1a/b cKO. We now provide evidence that DA axons from SNpc of Lmx1a/b cKO respond to Sema7a. Indeed, we found a significantly smaller amount of DA axons from SNpc of Lmx1a/b cKO embryos terminating on Sema7a stripes whereas DA axons from SNpc of control animals did not show any preference for Sema7a or IgG stripes (see supplementary Fig. 8)

2) Is the Otx2 expression in VTA and SNpc neurons of Lmx double KO mice co-localized with Plxnc1?

This is an interesting question but we could not perform double immunolabeling for Otx2 and Plxnc1 as our antibodies for these proteins are made in goat and sheep respectively and thus the secondary antibodies produce a lot of cross talk (anti-sheep recognises goat epitope and vice versa). We thus isolate Otx2+TH+ neurons in VTA and SNpc from Lmx1a/b cKO and we could validate that these neurons also express Plxnc1 (Fig. 7g-h). Inversely, with this method, we could not detect Plxnc1 from SNpc DA neurons of control mice.

3) Is the Plxnc1 expression maintained in VTA neurons of mice lacking Otx2? Indeed, based on the data here reported, it should be expected a phenotype mirroring that observed in Lmx double KO. In my opinion, with these improvements this manuscript should be published in Nature Communication.

We agree that it could have been interesting to verify if Plxnc1 is maintained in VTA of mice lacking Otx2. However, we could not obtain the conditional mutant mice for Otx2 to perform this experiment. We believe that it is beyond the scope of the present study. Although we did not perform this experiment, we downtone the statement that Otx2 maintain Plxnc1 expression in VTA. If Otx2 is required to Plxnc1 expression, we would anticipate a reduction or an absence of Plxnc1 in these animals. There are some evidences in the literature suggesting that Otx2 knockdown result in a reduction of Plxnc1. Chung et al., (Ref. 34) reported that ventral midbrain cell culture treated with a shRNA for Otx2 result in a small reduction of Plxnc1 whereas Otx2 overexpression results in a small increase in Plxnc1 expression. Although their results did not reach statistical significance, they are inline with our observations.

Reviewer #4 (Remarks to the Author):

The authors provide convincing data that Lmx1a/1b together with Otx2 determine levels of the axon guidance receptor Plxnc1 in the VTA dopamine neurons, and that these factors together with an additional unknown molecule control the levels of Plxnc1 in the SN as well. These combinations of factors result in high levels of Plxnc1 expression in the VTA and no expression in the SN. In the striatum, the axonal target area of mDA neurons, there is high expression of the Plxnc1 ligand Sema7a, restricted to the dorsal anterior striatum. They provide evidence that Sema7a functions as an inhibitory cue for VTA but not SN axons in their assay. They propose that that directs segregation of VTA axons and SN axons: VTA axons with high Plxnc1 avoid the Sema7a rich region, whereas SN axons innervate the Sema7a rich region.

The authors did a comprehensive set of studies including KO and overexpression of Lmx1a/b, KO of Sema7a, and examination of the response of neurons and explants to Sema7a in vitro.

The data are clear and convincing, and on the whole the paper is well written. It provides an additional set of cues that DA axons use to determine which region of the striatum they will innervate. The striking similarity of the phenotype in Lmx1a/1b KO mice to the previously reported Netrin1 KO mice is interesting and suggests that both sets of molecules play important roles.

I believe the paper needs a few minor textual changes indicated below. I also suggest an optional experiment that could enhance the interpretation.

1. Otx 2 appears to be a major player in the ultimate regulation of Plxnc1 expressed by VTA and SN neurons. Although well addressed in the discussion, it would be preferable also to mention Otx2 and its role in the abstract and introduction.

As suggested, we made changes in the introduction and in the abstract suggesting the role of Otx2 in the regulation of Plxnc1.

2. Since the difference in outgrowth of SN and VTA neurons in the presence of Sema7a in vitro did not differ very much using single neurons and Sholl analysis for quantification, it would be nice to see SN explants grown on stripes of Sema7a. It is expected that SN axons would ignore the boundary and grow equally well on both control and Sema 7a stripes. If not, the interpretation may be different.

As also suggested by reviewer 3, we performed stripes assays with SNpc explants (Supplementary Fig. 8). SNpc explants ignore the boundaries of Sema7a and grow equally on both control and Sema7a KO.

3. In wild-type animals, if Sema7a is preventing the VTA axons from innervating the dorsal striatum, what is preventing the SN axons from innervating the ventral striatum? It may be that as shown in ref 36, high Netrin-1 levels in ventral striatum prevents these axons from growing ventrally. Rather than wait and mention this study in the discussion it would seem more appropriate to cite in the introduction and discuss in the discussion (including discussing the Netrin1 KO phenotype, which currently is not discussed).

We agree with the reviewer and we now cited ref 36 and the netrin KO phenotype in the intro and discuss in more details these studies in the discussion.

4. The authors should cite other papers in this paragraph of the Introduction: "Very little is known about the molecules regulating axonal targeting of mDA neurons, and how the different mDA neuronal populations establish their specific connections is unknown." With proper attention to these minor points, the paper is appropriate for Nature Communications.

We agree and because of space limitation we decided to cite a good review article covering the subject.

REVIEWERS' COMMENTS:

Reviewer #1 (Remarks to the Author):

In this revised version of the manuscript, the authors have addressed most of the concerns I raised during the initial submission. In particular, they provide additional quantification of some analyses, performed clear back-labelling of tracts and performed additional timecourse studies that were essential to back-up their conclusions and claims.

In addition, they rewrote part of the text to clarify how they define dorsal and ventral striatal structures and the associated connectivity.

Overall, the revised version of the manuscript provides interesting molecular insights into a relatively unknown and functionally important process, the dopaminergic innervation of the striatum.

I thus now fully support publication.

Reviewer #2 (Remarks to the Author):

The authors have provided satisfactory responses to my questions. I find that the paper is now acceptable for publication.

Reviewer #3 (Remarks to the Author):

The manuscript by Chabrat et al. has been revised and in my opinion it is further improved.

As regarding to my specific comments, the authors provided partial experimental answers justifying why they have not performed the analysis of Otx2 CKO mutants.

Nevertheless, I think that in the present version the manuscript is acceptable for publication in Nature Communications.